# Strategic Distribution Shift of Interacting Agents via Coupled Gradient Flows

**Lauren Conger**
California Institute of Technology
`lconger@caltech.edu`

**Franca Hoffmann**
California Institute of Technology
`franca.hoffmann@caltech.edu`

**Eric Mazumdar**
California Institute of Technology
`mazumdar@caltech.edu`

**Lillian Ratliff**
University of Washington
`ratliffl@uw.edu`

## Abstract

We propose a novel framework for analyzing the dynamics of distribution shift in real-world systems that captures the feedback loop between learning algorithms and the distributions on which they are deployed. Prior work largely models feedback-induced distribution shift as adversarial or via an overly simplistic distribution-shift structure. In contrast, we propose a coupled partial differential equation model that captures fine-grained changes in the distribution over time by accounting for complex dynamics that arise due to strategic responses to algorithmic decision-making, non-local endogenous population interactions, and other exogenous sources of distribution shift. We consider two common settings in machine learning: cooperative settings with information asymmetries, and competitive settings where a learner faces strategic users. For both of these settings, when the algorithm retrains via gradient descent, we prove asymptotic convergence of the retraining procedure to a steady-state, both in finite and in infinite dimensions, obtaining explicit rates in terms of the model parameters. To do so we derive new results on the convergence of coupled PDEs that extends what is known on multi-species systems. Empirically, we show that our approach captures well-documented forms of distribution shifts like polarization and disparate impacts that simpler models cannot capture.

## 1 Introduction

In many machine learning tasks, there are commonly sources of exogenous and endogenous distribution shift, necessitating that the algorithm be retrained repeatedly over time. Some of these shifts occur without the influence of an algorithm; for example, individuals influence each other to become more or less similar in their attributes, or benign forms of distributional shift occur [Qui+]. Other shifts, however, are in response to algorithmic decision-making. Indeed, the very use of a decision-making algorithm can incentivize individuals to change or mis-report their data to achieve desired outcomes— a phenomenon known in economics as Goodhart's law. Such phenomena have been empirically observed, a well-known example being in [CC11], where researchers observed a population in Columbia strategically mis-reporting data to game a poverty index score used for distributing government assistance. Works such as [Mil+20; Wil+21], which investigate the effects of distribution shift over time on a machine learning algorithm, point toward the need for evaluating the robustness of algorithms to distribution shifts. Many existing approaches for modeling distribution shift focus on simple metrics like optimizing over moments or covariates [DY10; LHL21; BBS09]. Other methods consider worst-case scenarios, as in distributionally robust optimization [AZ22; LFG22; DN21; Kuh+19]. However, when humans respond to algorithms, these techniques may not

be sufficient to holistically capture the impact an algorithm has on a population. For example, an algorithm that takes into account shifts in a distribution's mean might inadvertently drive polarization, rendering a portion of the population disadvantaged.

Motivated by the need for a more descriptive model, we present an alternative perspective which allows us to fully capture complex dynamics that might drive distribution shifts in real-world systems. Our approach is general enough to capture various sources of exogenous and endogenous distribution shift including the feedback loop between algorithms and data distributions studied in the literature on performative prediction [Per+20; IYZ21; Ray+22; Nar+22; MPZ21], the strategic interactions studied in strategic classification [Har+16; Don+18], and also endogenous factors like intra-population dynamics and distributional shifts. Indeed, while previous works have studied these phenomena in isolation, our method allows us to capture all of them as well as their interactions. For example, in [Zrn+21], the authors investigate the effects of dynamics in strategic classification problems— but the model they analyze does not capture individual interactions in the population. In [IYZ21], the authors model the interaction between a population that repeatedly responds to algorithmic decision-making by shifting its mean. Additionally, [Ray+22] study settings in which the population has both exogenous and endogenous distribution shifts due to feedback, but much like the other cited work, the focus remains on average performance. Each of these works fails to account for diffusion or intra-population interactions that can result in important qualitative changes to the distribution.

**Contributions.** Our approach to this problem relies on a detailed non-local PDE model of the data distribution which captures each of these factors. One term driving the evolution of the distribution over time captures the response of the population to the deployed algorithm, another draws on models used in the PDE literature for describing non-local effects and consensus in biological systems to model intra-population dynamics, and the last captures a background source of distribution shift. This is coupled with an ODE, lifted to a PDE, which describes the training of a machine learning algorithm results in a coupled PDE system which we analyze to better understand the behaviors that can arise among these interactions.

In one subcase, our model exhibits a joint gradient flow structure, where both PDEs can be written as gradients flows with respect to the same joint energy, but considering infinite dimensional gradients with respect to the different arguments. This mathematical structure provides powerful tools for analysis and has been an emerging area of study with a relatively small body of prior work, none of which related to distribution shifts in societal systems, and a general theory for multi-species gradient flows is still lacking. We give a brief overview of the models that are known to exhibit this joint gradient flow structure: in [DS20] the authors consider a two-species tumor model with coupling through Brinkman's Law. A number of works consider coupling via convolution kernels [FF13; Giu+22; JPZ22; CHS18; DT20; Dou+23] and cross-diffusion [LY22; AB21; MKB14], with applications in chemotaxis among other areas. In the models we consider here, the way the interaction between the two populations manifests is neither via cross-diffusion, nor via the non-local self-interaction term. A related type of coupling has recently appeared in [HPS22; HPS23], however in the setting of graphs. Recent work [Dom+21] provides particle-based methods to approximately compute the solution to a minimax problem where the optimization space is over measures; following that work, [WC23] provides another particle-based method using mirror descent-ascent to solve a similar problem. Other recent work [Lu23] proves that a mean-field gradient ascent-descent scheme with an entropy annealing schedule converges to the solution of a minimax optimization problem with a timescale separation parameter that is also time-varying; in contrast, our work considers fixed timescale separation setting. [GG23] show that the mean-field description of a particle method for solving minimax problems has proveable convergence guarantees in the Wasserstein-Fisher-Rao metric. Each of these references considers an energy functional that is linear in the distribution of each species respectively; our energy includes nonlinearities in the distributions via a self-interaction term as well as diffusion for the population. Moreover, the above works introduce a gradient flow dynamic as a tool for obtaining and characterizing the corresponding steady states, whereas in our setting we seek to capture the time-varying behavior that models distributions shifts. In the other subcase, we prove exponential convergence in two competitive, timescale separated settings where the algorithm and strategic population have conflicting objectives. We show numerically that retraining in a competitive setting leads to polarization in the population, illustrating the importance of fine-grained modeling.

## 2 Problem Formulation

Machine learning algorithms that are deployed into the real world for decision-making often become part of complex feedback loops with the data distributions and data sources with which they interact. In an effort to model these interactions, consider a machine learning algorithm that has loss given by $L(z, x)$ where $x \in \mathbb{R}^d$ are the algorithm parameters and $z \in \mathbb{R}^d$ are the population attributes, and the goal is to solve

$$\underset{x \in \mathcal{X}}{\operatorname{argmin}} \underset{z \sim \rho}{\mathbb{E}} L(z, x),$$

where $\mathcal{X}$ is the class of model parameters and $\rho(z)$ is the population distribution. Individuals have an objective given by $J(z, x)$ in response to a model parameterized by $x$, and they seek to solve

$$\underset{z \in \mathbb{R}^d}{\operatorname{argmin}} J(z, x).$$

When individuals in the population and the algorithm have access to gradients, we model the optimization process as a gradient-descent-type process. Realistically, individuals in the population will have nonlocal information and influences, as well as external perturbations, the effects of which we seek to capture in addition to just minimization. To address this, we propose a partial differential equation (PDE) model for the population, that is able to capture nonlocal interactions between individuals on the level of a collective population. To analyse how the population evolves over time, a notion of derivative in infinite dimensions is needed. A natural, and in this context physically meaningful, way of measuring the dissipation mechanism for probability distributions is the Wasserstein-2 metric (see Definition 4). The following expression appears when computing the gradient of an energy functional with respect to the Wasserstein-2 topology.

**Definition 1.** *[First Variation] For a map $G : \mathcal{P}(\mathbb{R}^d) \mapsto \mathbb{R}$ and fixed probability distribution $\rho \in \mathcal{P}(\mathbb{R}^d)$, the* first variation *of $G$ at the point $\rho$ is denoted by $\delta_\rho G[\rho] : \mathbb{R}^d \to \mathbb{R}$, and is defined via the relation*

$$\int \delta_\rho G[\rho](z)\psi(z)\mathrm{d}z = \lim_{\epsilon \to 0} \frac{1}{\epsilon}(G(\rho + \epsilon\psi) - G(\rho))$$

*for all $\psi \in C_c^\infty(\mathbb{R}^d)$ such that $\int \mathrm{d}\psi = 0$, assuming that $G$ is regular enough for all quantities to exist.*

Here, $\mathcal{P}(\mathbb{R}^d)$ denotes the space of probability measures on the Borel sigma algebra. Using the first variation, we can express the gradient in Wasserstein-2 space, see for example [Vil03, Exercise 8.8].

**Lemma 1.** *The gradient of an energy $G : \mathcal{P}_2(\mathbb{R}^d) \to \mathbb{R}$ in the Wasserstein-2 space is given by*

$$\nabla_{W_2} G(\rho) = -\operatorname{div}\left(\rho\nabla\delta_\rho G[\rho]\right).$$

Here, $\mathcal{P}_2(\mathbb{R}^d)$ denotes the set of probability measures with bounded second moments, also see Appendix A.2. As a consequence, the infinite dimensional steepest descent in Wasserstein-2 space can be expressed as the PDE

$$\partial_t \rho = -\nabla_{W_2} G(\rho) = \operatorname{div}\left(\rho\nabla\delta_\rho G[\rho]\right). \tag{1}$$

All the coupled gradient flows considered in this work have this Wasserstein-2 structure. In particular, when considering that individuals minimize their own loss, we can capture these dynamics via a gradient flow in the Wasserstein-2 metric on the level of the distribution of the population. Then for given algorithm parameters $x \in \mathbb{R}^d$, the evolution for this strategic population is given by

$$\partial_t \rho = \operatorname{div}\left(\rho\nabla\delta_\rho\Big[\underset{z \sim \rho}{\mathbb{E}} J(z, x) + E(\rho)\Big]\right), \tag{2}$$

where $E(\rho)$ is a functional including terms for internal influences and external perturbations. In real-world deployment of algorithms, decision makers update their algorithm over time, leading to an interaction between the two processes. We also consider the algorithm dynamics over time, which we model as

$$\dot{x} = -\nabla_x\Big[\underset{z \sim \rho}{\mathbb{E}} L(z, x)\Big]. \tag{3}$$

In this work, we analyze the behavior of the dynamics under the following model. The algorithm suffers a cost $f_1(z,x)$ for a data point $z$ under model parameters $x$ in the strategic population, and a cost $f_2(z,x)$ for a data point in a fixed, non-strategic population. The strategic population is denoted by $\rho \in \mathcal{P}$, and the non-strategic population by $\bar{\rho} \in \mathcal{P}$. The algorithm aims to minimize

$$\mathbb{E}_{z \sim \rho} L(z,x) = \int f_1(z,x)\mathrm{d}\rho(z) + \int f_2(z,x)\mathrm{d}\bar{\rho}(z) + \frac{\beta}{2} \|x - x_0\|^2 ,$$

where the norm is the vector inner product $\|x\|^2 = \langle x, x \rangle$ and $\beta > 0$ weights the cost of updating the model parameters from its initial condition.

We consider two settings: $(i)$ aligned objectives, and $(ii)$ competing objectives. Case $(i)$ captures the setting in which the strategic population minimization improves the performance of the algorithm, subject to a cost for deviating from a reference distribution $\tilde{\rho} \in \mathcal{P}$. This cost stems from effort required to manipulate features, such as a loan applicant adding or closing credit cards. On the other hand, Case $(ii)$ captures the setting in which the strategic population minimization worsens the performance of the algorithm, again incurring cost from distributional changes.

## 2.1 Case (i): Aligned Objectives

In this setting, we consider the case where the strategic population and the algorithm have aligned objectives. This occurs in examples such as recommendation systems, where users and algorithm designers both seek to develop accurate recommendations for the users. This corresponds to the population cost

$$\mathbb{E}_{z \sim \rho, x \sim \mu} J(z,x) = \iint f_1(z,x)\mathrm{d}\rho(z)\mathrm{d}\mu(x) + \alpha KL(\rho \,|\, \tilde{\rho}),$$

where $KL(\cdot \,|\, \cdot)$ denotes the Kullback-Leibler divergence. Note that the KL divergence introduces diffusion to the dynamics for $\rho$. The weight $\alpha > 0$ parameterizes the cost of distribution shift to the population. To account for nonlocal information and influence among members of the population, we include a kernel term $E(\rho) = \frac{1}{2} \int \rho W * \rho \,\mathrm{d}z$, where $(W * \rho)(z) = \int W(z - \bar{z})\mathrm{d}\rho(\bar{z})$ is a convolution integral and $W$ is a suitable interaction potential.

## 2.2 Case (ii): Competing Objectives

In settings such as online internet forums, where algorithms and users have used manipulative strategies for marketing [Del06], the strategic population may be incentivized to modify or mis-report their attributes. The algorithm has a competitive objective, in that it aims to maintain performance against a population whose dynamics cause the algorithm performance to suffer. When the strategic population seeks an outcome contrary to the algorithm, we model strategic population cost as

$$\mathbb{E}_{z \sim \rho, x \sim \mu} J(z,x) = - \iint f_1(z,x)\mathrm{d}\rho(z)\mathrm{d}\mu(x) + \alpha KL(\rho \,|\, \tilde{\rho}).$$

A significant factor in the dynamics for the strategic population is the timescale separation between the two "species"—i.e., the population and the algorithm. In our analysis, we will consider two cases: one, where the population responds much faster than the algorithm, and two, where the algorithm responds much faster than the population. We illustrate the intermediate case in a simulation example.

## 3 Results

We are interested in characterizing the long-time asymptotic behavior of the population distribution, as it depends on the decision-makers action over time. The structure of the population distribution gives us insights about how the decision-makers actions influences the entire population of users. For instance, as noted in the preceding sections, different behaviors such as bimodal distributions or large tails or variance might emerge, and such effects are not captured in simply looking at average performance. To understand this intricate interplay, one would like to characterize the behavior of both the population and the algorithm over large times. Our main contribution towards this goal is a novel analytical framework as well as analysis of the long-time asymptotics.

A key observation is that the dynamics in (2) and (3) can be re-formulated as a gradient flow; we lift $x$ to a probability distribution $\mu$ by representing it as a Dirac delta $\mu$ sitting at the point $x$. As a result, the evolution of $\mu$ will be governed by a PDE, and combined with the PDE for the population, we obtain a system of coupled PDEs,

$$\partial_t \rho = \text{div}\left(\rho \nabla_z \delta_\rho \big[\mathop{\mathbb{E}}_{z \sim \rho, x \sim \mu} J(z, x) + E(\rho)\big]\right)$$

$$\partial_t \mu = \text{div}\left(\mu \nabla_x \delta_\mu \big[\mathop{\mathbb{E}}_{z \sim \rho, x \sim \mu} L(z, x)\big]\right),$$

where $\delta_\rho$ and $\delta_\mu$ are first variations with respect to $\rho$ and $\mu$ according to Definition 1. The natural candidates for the asymptotic profiles of this coupled system are its steady states, which - thanks to the gradient flow structure - can be characterized as ground states of the corresponding energy functionals. In this work, we show existence and uniqueness of minimizers (maximizers) for the functionals under suitable conditions on the dynamics. We also provide criteria for convergence and explicit convergence rates. We begin with the case where the interests of the population and algorithm are aligned, and follow with analogous results in the competitive setting. We show convergence in energy, which in turn ensures convergence in a product Wasserstein metric. For convergence in energy, we use the notion of relative energy and prove that the relative energy converges to zero as time increases.

**Definition 2** (Relative Energy). *The relative energy of a functional $G$ is given by $G(\gamma|\gamma_\infty) = G(\gamma) - G(\gamma_\infty)$, where $G(\gamma_\infty)$ is the energy at the steady state.*

Since we consider the joint evolution of two probability distributions, we define a distance metric $\overline{W}$ on the product space of probability measures with bounded second moment.

**Definition 3** (Joint Wasserstein Metric). *The metric over $\mathcal{P}_2(\mathbb{R}^d) \times \mathcal{P}_2(\mathbb{R}^d)$ is called $\overline{W}$ and is given by*

$$\overline{W}((\rho, \mu), (\tilde{\rho}, \tilde{\mu}))^2 = W_2(\rho, \tilde{\rho})^2 + W_2(\mu, \tilde{\mu})^2$$

*for all pairs $(\rho, \mu), (\tilde{\rho}, \tilde{\mu}) \in \mathcal{P}_2(\mathbb{R}^d) \times \mathcal{P}_2(\mathbb{R}^d)$, and where $W_2$ denotes the Wasserstein-2 metric (see Definition 4). We denote by $\overline{\mathcal{W}}(\mathbb{R}^d) := (\mathcal{P}_2(\mathbb{R}^d) \times \mathcal{P}_2(\mathbb{R}^d), \overline{W})$ the corresponding metric space.*

## 3.1 Gradient Flow Structure

In the case where the objectives of the algorithm and population are *aligned*, we can write the dynamics as a gradient flow by using the same energy functional for both species. Let $G_a(\rho, \mu) : \mathcal{P}(\mathbb{R}^d) \times \mathcal{P}(\mathbb{R}^d) \mapsto [0, \infty]$ be the energy functional given by

$$G_a(\rho, \mu) = \iint f_1(z, x)\mathrm{d}\rho(z)\mathrm{d}\mu(x) + \iint f_2(z, x)\mathrm{d}\bar{\rho}(z)\mathrm{d}\mu(x) + \alpha KL(\rho|\tilde{\rho}) + \frac{1}{2}\int \rho W * \rho$$
$$+ \frac{\beta}{2}\int \|x - x_0\|^2 \, \mathrm{d}\mu(x).$$

This expression is well-defined as the relative entropy $KL(\rho \,|\, \tilde{\rho})$ can be extended to the full set $\mathcal{P}(\mathbb{R}^d)$ by setting $G_a(\rho, \mu) = +\infty$ in case $\rho$ is not absolutely continuous with respect to $\tilde{\rho}$.

In the *competitive* case we define $G_c(\rho, x) : \mathcal{P}(\mathbb{R}^d) \times \mathbb{R}^d \mapsto [-\infty, \infty]$ by

$$G_c(\rho, x) = \int f_1(z, x)\mathrm{d}\rho(z) + \int f_2(x, z')\mathrm{d}\bar{\rho}(z') - \alpha KL(\rho|\tilde{\rho}) - \frac{1}{2}\int \rho W * \rho + \frac{\beta}{2}\|x - x_0\|^2.$$

In settings like recommender systems, the population and algorithm have aligned objectives; they seek to minimize the same cost but are subject to different dynamic constraints and influences, modeled by the regularizer and convolution terms. In the case where the objectives are aligned, the dynamics are given by

$$\partial_t \rho = \text{div}\left(\rho \nabla_z \delta_\rho G_a[\rho, \mu]\right)$$
$$\partial_t \mu = \text{div}\left(\mu \nabla_x \delta_\mu G_a[\rho, \mu]\right). \tag{4}$$

Note that (4) is a joint gradient flow, because the dynamics can be written in the form

$$\partial_t \gamma = \text{div}\left(\gamma \nabla \delta_\gamma G_a(\gamma)\right),$$

where $\gamma = (\rho, \mu)$ and where the gradient and divergence are taken in both variables $(z, x)$. We discuss the structure of the dynamics (4) as well as the meaning of the different terms appearing in the energy functional $G_a$ in Appendix A.1.

In other settings, such as credit score reporting, the objectives of the population are competitive with respect to the algorithm. Here we consider two scenarios; one, where the algorithm responds quickly relative to the population, and two, where the population responds quickly relative to the algorithm. In the case where the algorithm can immediately adjust optimally (best-respond) to the distribution, the dynamics are given by

$$\partial_t \rho = -\text{div}\left(\rho\left(\nabla_z \delta_\rho G_c[\rho, x]\right)|_{x=b(\rho)}\right),$$
$$b(\rho) := \underset{\bar{x}}{\text{argmin}}\, G_c(\rho, \bar{x}). \tag{5}$$

Next we can consider the population immediately responding to the algorithm, which has dynamics

$$\frac{\mathrm{d}}{\mathrm{d}t} x = -\nabla_x G_c(\rho, x)|_{\rho=r(x)},$$
$$r(x) := \underset{\hat{\rho}\in\mathcal{P}}{\text{argmin}} -G_c(\hat{\rho}, x). \tag{6}$$

In this time-scale separated setting, model (5) represents a dyamic maximization of $G_c$ with respect to $\rho$ in Wasserstein-2 space, and an instantaneous minimization of $G_c$ with respect to the algorithm parameters $x$. Model (6) represents an instantaneous maximization of $G_c$ with respect to $\rho$ and a dynamic minimization of $G_c$ with respect to the algorithm parameters $x$. The key results on existence and uniqueness of a ground state as well as the convergence behavior of solutions depend on convexity (concavity) of $G_a$ and $G_c$. The notion of convexity that we will employ for energy functionals in the Wasserstein-2 geometry is *(uniform) displacement convexity*, which is analogous to (strong) convexity in Euclidean spaces. One can think of displacement convexity for an energy functional defined on $\mathcal{P}_2$ as convexity along the shortest path in the Wasserstein-2 metric (linear interpolation in the Wasserstein-2 space) between any two given probability distributions. For a detailed definition of (uniform) displacement convexity and concavity, see Section A.2. In fact, suitable convexity properties of the input functions $f_1, f_2, W$ and $\tilde{\rho}$ will ensure (uniform) displacement convexity of the resulting energy functionals appearing in the gradient flow structure, see for instance [Vil03, Chapter 5.2].

We make the following assumptions in both the competitive case and aligned interest cases. Here, $\text{I}_d$ denotes the $d \times d$ identity matrix, Hess $(f)$ denotes the Hessian of $f$ in all variables, while $\nabla_x^2 f$ denotes the Hessian of $f$ in the variable $x$ only.

**Assumption 1** (Convexity of $f_1$ and $f_2$). *The functions $f_1, f_2 \in C^2(\mathbb{R}^d \times \mathbb{R}^d; [0, \infty))$ satisfy for all $(z, x) \in \mathbb{R}^d \times \mathbb{R}^d$ the following:*

- *There exists constants $\lambda_1, \lambda_2 \geq 0$ such that Hess $(f_1) \succeq \lambda_1 \text{I}_{2d}$ and $\nabla_x^2 f_2 \succeq \lambda_2 \text{I}_d$;*
- *There exist constants $a_i > 0$ such that $x \cdot \nabla_x f_i(z, x) \geq -a_i$ for $i = 1, 2$;*

**Assumption 2** (Reference Distribution Shape). *The reference distribution $\tilde{\rho} \in \mathcal{P}(\mathbb{R}^d) \cap L^1(\mathbb{R}^d)$ satisfies $\log \tilde{\rho} \in C^2(\mathbb{R}^d)$ and $\nabla_z^2 \log \tilde{\rho}(z) \preceq -\tilde{\lambda} \text{I}_d$ for some $\tilde{\lambda} > 0$.*

**Assumption 3** (Convex Interaction Kernel). *The interaction kernel $W \in C^2(\mathbb{R}^d; [0, \infty))$ is convex, symmetric $W(-z) = W(z)$, and for some $D > 0$ satisfies*

$$z \cdot \nabla_z W(z) \geq -D, \quad |\nabla_z W(z)| \leq D(1 + |z|) \quad \forall z \in \mathbb{R}^d.$$

We make the following observations regarding the assumptions above:

- The convexity in Assumption 3 can be relaxed and without affecting the results outlined below by following a more detailed analysis analogous to the approach in [CMV03].
- If $f_1$ and $f_2$ are strongly convex, the proveable convergence rate increases, but without strict or strong convexity of $f_1$ and $f_2$, the regularizers $KL(\rho|\tilde{\rho})$ and $\int \|x - x_0\|_2^2 \,\mathrm{d}x$ provide the convexity guarantees necessary for convergence.

For concreteness, one can consider the following classical choices of input functions to the evolution:

- Using the log-loss function for $f_1$ and $f_2$ satisfies Assumption 1.

- Taking the reference measure $\tilde{\rho}$ to be the normal distribution satisfies Assumption 2, which ensures the distribution is not too flat.
- Taking quadratic interactions $W(z) = \frac{1}{2}|z|^2$ satisfies Assumption 3.

**Remark 1** (Cauchy-Problem). *To complete the arguments on convergence to equilibrium, we require sufficient regularity of solutions to the PDEs under consideration. In fact, it is sufficient if we can show that equations (4), (5), and (6) can be approximated by equations with smooth solutions. Albeit tedious, these are standard techniques in the regularity theory for partial differential equations, see for example [CMV03, Proposition 2.1 and Appendix A], [OV00], [Vil03, Chapter 9], and the references therein. Similar arguments as in [DV00] are expected to apply to the coupled gradient flows considered here, guaranteeing existence of smooth solutions with fast enough decay at infinity, and we leave a detailed proof for future work.*

## 3.2 Analysis of Case (i): Aligned Objectives

The primary technical contribution of this setting consists of lifting the algorithm dynamics from an ODE to a PDE, which allows us to model the system as a joint gradient flow on the product space of probability measures. The coupling occurs in the potential function, rather than as cross-diffusion or non-local interaction as more commonly seen in the literature for multi-species systems.

**Theorem 2.** *Suppose that Assumptions 1-3 are satisfied and let $\lambda_a := \lambda_1 + \min(\lambda_2 + \beta, \alpha\tilde{\lambda}) > 0$. Consider solutions $\gamma_t := (\rho_t, \mu_t)$ to the dynamics (4) with initial conditions satisfying $\gamma_0 \in \mathcal{P}_2(\mathbb{R}^d) \times \mathcal{P}_2(\mathbb{R}^d)$ and $G_a(\gamma_0) < \infty$. Then the following hold:*

*(a) There exists a unique minimizer $\gamma_\infty = (\rho_\infty, \mu_\infty)$ of $G_a$, which is also a steady state for equation (4). Moreover, $\rho_\infty \in L^1(\mathbb{R}^d)$, has the same support as $\tilde{\rho}$, and its density is continuous.*

*(b) The solution $\gamma_t$ converges exponentially fast in $G_a(\cdot \mid \gamma_\infty)$ and $\overline{W}$,*
$$G_a(\gamma_t \mid \gamma_\infty) \leq e^{-2\lambda_a t} G_a(\gamma_0 \mid \gamma_\infty) \quad \text{and} \quad \overline{W}(\gamma_t, \gamma_\infty) \leq ce^{-\lambda_a t} \quad \text{for all } t \geq 0\,,$$
*where $c > 0$ is a constant only depending on $\gamma_0$, $\gamma_\infty$ and the parameter $\lambda_a$.*

*Proof.* (Sketch) For existence and uniqueness, we leverage classical techniques in the calculus of variations. To obtain convergence to equilibrium in energy, our key result is a new HWI-type inequality, providing as a consequence generalizations of the log-Sobolev inequality and the Talagrand inequality. Together, these inequalities relate the energy (classically denoted by $H$ in the case of the Boltzmann entropy), the metric (classically denoted by $W$ in the case of the Wasserstein-2 metric) and the energy dissipation (classically denoted by $I$ in the case of the Fisher information)[1]. Combining these inequalities with Gronwall's inequality allows us to deduce convergence both in energy and in the metric $\overline{W}$. □

## 3.3 Analysis of Case (ii): Competing Objectives

In this setting, we consider the case where the algorithm and the strategic population have goals in opposition to each other; specifically, the population benefits from being classified incorrectly. First, we will show that when the algorithm instantly best-responds to the population, then the distribution of the population converges exponentially in energy and in $W_2$. Then we will show a similar result for the case where the population instantly best-responds to the algorithm.

In both cases, we begin by proving two Danskin-type results (see [Dan67; Ber71]) which will be used for the main convergence theorem, including convexity (concavity) results. To this end, we make the following assumption ensuring that the regularizing component in the evolution of $\rho$ is able to control the concavity introduced by $f_1$ and $f_2$.

**Assumption 4** (Upper bounds for $f_1$ and $f_2$). *There exists a constant $\Lambda_1 > 0$ such that*
$$\nabla_z^2 f_1(z, x) \preceq \Lambda_1 I_d \qquad \text{for all } (z, x) \in \mathbb{R}^d \times \mathbb{R}^d\,,$$
*and for any $R > 0$ there exists a constant $c_2 = c_2(R) \in \mathbb{R}$ such that*
$$\sup_{x \in B_R(0)} \int f_2(z, x) \mathrm{d}\bar{\rho}(z) < c_2\,.$$

---

[1] Hence the name HWI inequalities.

Equipped with Assumption 4, we state the result for a best-responding algorithm.

**Theorem 3.** *Suppose Assumptions 1-4 are satisfied with $\alpha\tilde{\lambda} > \Lambda_1$. Let $\lambda_b \coloneqq \alpha\tilde{\lambda} - \Lambda_1$. Define $G_b(\rho) \coloneqq G_c(\rho, b(\rho))$. Consider a solution $\rho_t$ to the dynamics (5) with initial condition $\rho_0 \in \mathcal{P}_2(\mathbb{R}^d)$ such that $G_b(\rho_0) < \infty$. Then the following hold:*

    *(a) There exists a unique maximizer $\rho_\infty$ of $G_b(\rho)$, which is also a steady state for equation (5). Moreover, $\rho_\infty \in L^1(\mathbb{R}^d)$, has the same support as $\tilde{\rho}$, and its density is continuous.*

    *(b) The solution $\rho_t$ converges exponentially fast to $\rho_\infty$ with rate $\lambda_b$ in $G_b(\cdot \mid \rho_\infty)$ and $W_2$,*

$$G_b(\rho_t \mid \rho_\infty) \leq e^{-2\lambda_b t} G_a(\rho_0 \mid \rho_\infty) \quad and \quad W_2(\rho_t, \rho_\infty) \leq ce^{-\lambda_b t} \quad for\ all\ t \geq 0\,,$$

    *where $c > 0$ is a constant only depending on $\rho_0$, $\rho_\infty$ and the parameter $\lambda_b$.*

*Proof.* (Sketch) The key addition in this setting as compared with Theorem 2 is proving that $G_b(\rho)$ is bounded below, uniformly displacement concave and guaranteeing its smoothness via Berge's Maximum Theorem. This is non-trivial as it uses the properties of the best response $b(\rho)$. A central observation for our arguments to work is that $\delta_\rho G_b[\rho] = (\delta_\rho G_c[\rho, x])\,|_{x=b(\rho)}$. We can then conclude using the direct method in the calculus of variations and the HWI method. □

Here, the condition that $\alpha\tilde{\lambda}$ must be large enough corresponds to the statement that the system must be subjected to a strong enough regularizing effect.

In the opposite case, where $\rho$ instantly best-responds to the algorithm, we show Danskin-like results for derivatives through the best response function and convexity of the resulting energy in $x$ which allows to deduce convergence.

**Theorem 4.** *Suppose Assumptions 1-4 are satisfied with $\alpha\tilde{\lambda} > \Lambda_1$, and that $r(x)$ is differentiable (as shown by example conditions in Lemmas 27 and 28). Define $G_d(x) \coloneqq G_c(r(x), x)$. Then it holds:*

    *(a) There exists a unique minimizer $x_\infty$ of $G_d(x)$ which is also a steady state for (6).*

    *(b) The vector $x(t)$ solving the dynamics (6) with initial condition $x(0) \in \mathbb{R}^d$ converges exponentially fast to $x_\infty$ with rate $\lambda_d \coloneqq \lambda_1 + \lambda_2 + \beta > 0$ in $G_d$ and in the Euclidean norm:*

$$\|x(t) - x_\infty\| \leq e^{-\lambda_d t}\|x(0) - x_\infty\|\,,$$
$$G_d(x(t)) - G_d(x_\infty) \leq e^{-2\lambda_d t}\left(G_d(x(0)) - G_d(x_\infty)\right)$$

    *for all $t \geq 0$.*

These two theorems illustrate that, under sufficient convexity conditions on the cost functions, we expect the distribution $\rho$ and the algorithm $x$ to converge to a steady state. In practice, when the distributions are close enough to the steady state there is no need to retrain the algorithm.

While we have proven results for the extreme timescale cases, we anticipate convergence to the same equilibrium in the intermediate cases. Indeed, it is well known [Bor09] (especially for systems in Euclidean space) that for two-timescale stochastic approximations of dynamical systems, with appropriate stepsize choices, converge asymptotically, and finite-time high probability concentration bounds can also be obtained. These results have been leveraged in strategic classification [Zrn+21] and Stackelberg games [FCR20; FR21; Fie+21]. We leave this intricate analysis to future work.

In the following section we show numerical results in the case of a best-responding $x$, best-responding $\rho$, and in between where $x$ and $\rho$ evolve on a similar timescale. Note that in these settings, the dynamics do not have a gradient flow structure due to a sign difference in the energies, requiring conditions to ensure that one species does not dominate the other.

## 4 Numerical Examples

We illustrate numerical results for the case of a classifier, which are used in scenarios such as loan or government aid applications [CC11], school admissions [PS13], residency match [Ree18], and recommendation algorithms [LSW10], all of which have some population which is incentivized to submit data that will result in a desirable classification. For all examples, we select classifiers

of the form $x \in \mathbb{R}$, so that a data point $z \in \mathbb{R}$ is assigned a label of 1 with probability $q(z, x) = (1 + \exp(-b^\top z + x))^{-1}$ where $b > 0$. Let $f_1$ and $f_2$ be given by

$$f_1(z, x) = -\log(1 - q(z, x)), \qquad f_2(z, x) = -\log q(z, x).$$

Note that $\mathrm{Hess}(f_1) \succeq 0$ and $\nabla_x^2 f_2 \succeq 0$, so $\lambda_1 = \lambda_2 = 0$. Here, the strictness of the convexity of the functional is coming from the regularizers, not the cost functions, with $\tilde{\rho}$ a scaled normal distribution. We show numerical results for two scenarios with additional settings in the appendix. First we illustrate competitive interests under three different timescale settings. Then we simulate the classifier taking an even more naïve strategy than gradient descent and discuss the results. The PDEs were implemented based on the finite volume method from [CCH15].

## 4.1 Competitive Objectives

In the setting with competitive objectives, we utilize $G_c(\rho, x)$ with $W = 0$, $f_1$ and $f_2$ as defined above with $b = 3$ fixed as it only changes the steepness of the classifier for $d = 1$, and $\alpha = 0.1$ and $\beta = 0.05$. In Figure 1, we simulate two extremes of the timescale setting; first when $\rho$ is nearly best-responding and then when $x$ is best-responding. The simulations have the same initial conditions and end with the same distribution shape; however, the behavior of the strategic population differs in the intermediate stages. When $\rho$ is nearly best-responding, we see that the distribution quickly shifts

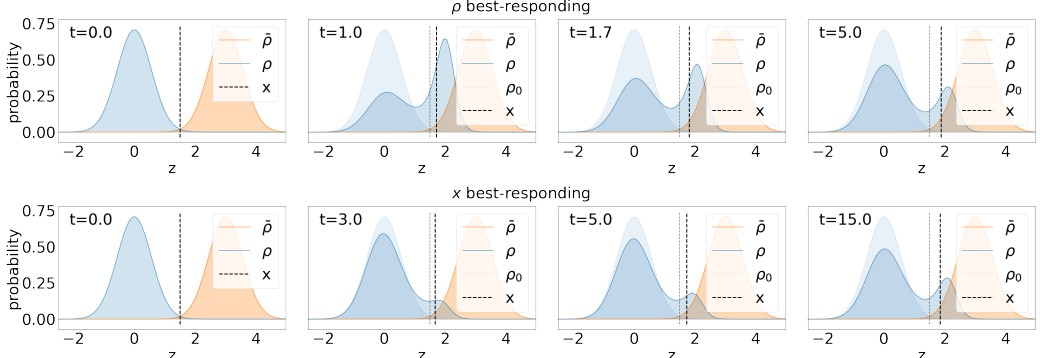

Figure 1: When $x$ versus $\rho$ best-responds, we observe the same final state but different intermediate states. Modes appear in the strategic population which simpler models cannot capture.

mass over the classifier threshold. Then the classifier shifts right, correcting for the shift in $\rho$, which then incentivizes $\rho$ to shift more mass back to the original mode. In contrast, when $x$ best-responds, the right-hand mode slowly increases in size until the system converges.

Figure 2 shows simulation results from the setting where $\rho$ and $x$ evolve on the same timescale. We observe that the distribution shift in $\rho$ appears to fall between the two extreme timescale cases, which we expect. We highlight two important observations for the competitive case. One, a single-mode

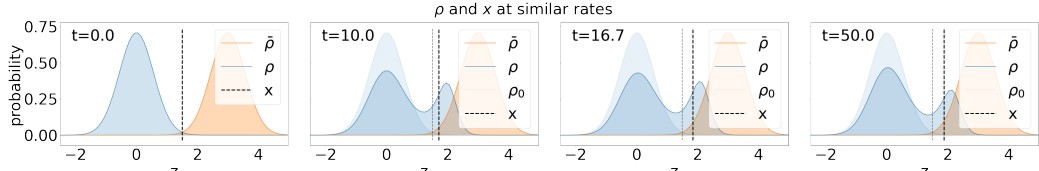

Figure 2: In this experiment the population and classifier have similar rates of change, and the distribution change for $\rho$ exhibits behaviors from both the fast $\rho$ and fast $x$ simulations; the right-hand mode does not peak as high as the fast $\rho$ case but does exceed its final height and return to the equilibrium.

distribution becomes bimodel, which would not be captured using simplistic metrics such as the mean and variance. This split can be seen as polarization in the population, a phenomenon that a mean-based strategic classification model would not capture. Two, the timescale on which the

classifier updates significantly impacts the intermediate behavior of the distribution. In our example, when $x$ updated slowly relative to the strategic population, the shifts in the population were greater than in the other two cases. This suggests that understanding the effects of timescale separation are important for minimizing volatility of the coupled dynamics.

## 4.2 Naïve Behavior

In this example, we explore the results of the classifier adopting a non-gradient-flow strategy, where the classifier chooses an initially-suboptimal value for $x$ and does not move, allowing the strategic population to respond. All functions and parameters are the same as in the previous example. When

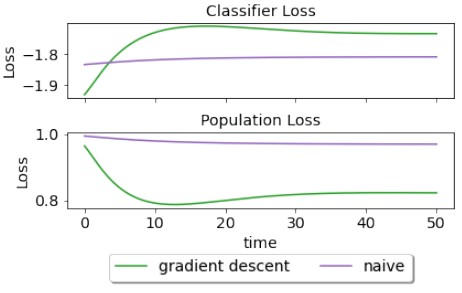

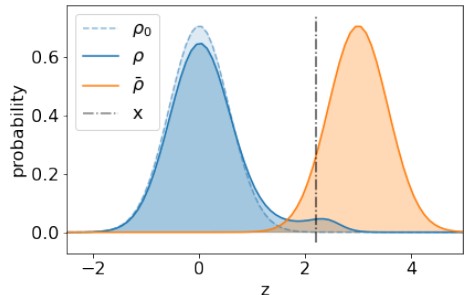

(a) Both species minimize their respective losses; when the classifier uses a naïve strategy, the final performance is better for the classifier and uniformly worse for the population.

(b) The classifier selects a suboptimal initial condition $x = 2.2$, instead of $x = 1.5$ which minimizes the initial loss, and then does not move in response to the population.

Figure 3: Although the classifier starts with a larger cost by taking the naive strategy, the final loss is better. This illustrates how our model can be used to compare robustness of different strategies against a strategic population.

comparing with the gradient descent strategy, we observe that while the initial loss for the classifier is worse for the naive strategy, the final cost is better. While this results is not surprising, because one can view this as a general-sum game where the best response to a fixed decision may be better than the equilibrium, it illustrates how our method provides a framework for evaluating how different training strategies perform in the long run against a strategic population.

## 5 Future Directions, Limitations, and Broader Impact

Our work presents a method for evaluating the robustness of an algorithm to a strategic population, and investigating a variety of robustness using our techniques opens a range of future research directions. Our application suggests many questions relevant to the PDE literature, such as: (1) Does convergence still hold with the gradient replaced by an estimated gradient? (2) Can we prove convergence in between the two timescale extremes? (3) How do multiple dynamic populations respond to an algorithm, or multiple algorithms? In the realm of learning algorithms, our framework can be extended to other learning update strategies and presents a way to model how we can design these update strategies to induce desired behaviors in the population.

A challenge in our method is that numerically solving high-dimensional PDEs is computationally expensive and possibly unfeasible. Here we note that in many applications, agents in the population do not alter more than a few features due to the cost of manipulation. We are encouraged by the recent progress using deep learning to solve PDEs, which could be used in our application.

**Broader Impacts** Modeling the full population distribution rather than simple metrics of the distribution is important because not all individuals are affected by the algorithm in the same way. For example, if there are tails of the distribution that have poor performance even if on average the model is good, we need to know how that group is advantaged or disadvantaged relative to the rest of the population. Additionally, understanding how people respond to algorithms offers an opportunity to incentivise people to move in a direction that increases social welfare.

## Acknowledgments and Disclosure of Funding

LC is supported by an NDSEG fellowhip from the Air Force Office of Scientific Research. FH is supported by start-up funds at the California Institute of Technology. LR is supported by ONR YIP N00014-20-1-2571 P00003 and NSF Awards CAREER 1844729, and CPS 1931718. EM acknowledges support from NSF Award 2240110. We are grateful for helpful discussions with José A. Carrillo.

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

# A  General structure and preliminaries

In this section, we give more details on the models discussed in the main article, and introduce definitions and notation that are needed for the subsequent proofs.

## A.1  Structure of the dynamics

For the case of aligned objectives, the full coupled system of PDEs (4) can be written as

$$\partial_t \rho = \alpha \Delta \rho + \text{div}\left(\rho \nabla_z \left(\int f_1 \mathrm{d}\mu - \alpha \log \tilde{\rho} + W * \rho\right)\right), \tag{7a}$$

$$\partial_t \mu = \text{div}\left(\mu \nabla_x \left(\int f_1 \mathrm{d}\rho + \int f_2 \mathrm{d}\bar{\rho} + \frac{\beta}{2}\|x - x_0\|^2\right)\right). \tag{7b}$$

In other words, the population $\rho$ in (7a) is subject to an isotropic diffusive force with diffusion coefficient $\alpha > 0$, a drift force due to the time-varying confining potential $\int f_1 \mathrm{d}\mu(t) - \alpha \log \tilde{\rho}$, and a self-interaction force via the interaction potential $W$. If we consider the measure $\mu$ to be given and fixed in time, this corresponds exactly to the type of parabolic equation studied in [CMV03]. Here however the dynamics are more complex due to the coupling of the confining potential with the dynamics (7b) for $\mu(t)$ via the coupling potential $f_1$. Before presenting the analysis of this model, let us give a bit more intuition on the meaning and the structure of these dynamics.

In the setting where $\mu$ represents a binary classifier, we can think of the distribution $\bar{\rho}$ as modelling all those individuals carrying the true label 1, say, and the distribution $\rho(t)$ as modelling all those individuals carrying a true label 0, say, where 0 and 1 denote the labels of two classes of interest. The term $\int f_1(z,x)\mu(t, \mathrm{d}x)$ represents a penalty for incorrectly classifying an individual at $z$ with true label 0 when using the classifier $\mu(t, x)$. In other words, $\int f_1(z,x)\mu(t, \mathrm{d}x) \in [0, \infty)$ is increasingly large the more $z$ digresses from the correct classification 0. Similarly, $\int f_1(z,x)\rho(t, \mathrm{d}z) \in [0, \infty)$ is increasingly large if the population $\rho$ shifts mass to locations in $z$ where the classification is incorrect. The terminology *aligned objectives* refers to the fact that in (7) both the population and the classifier are trying to evolve in a way as to maximize correct classification. Analogously, the term $\int f_2(z,x)\bar{\rho}(\mathrm{d}z)$ is large if $x$ would incorrectly classify the population $\bar{\rho}$ that carries the label 1. A natural extension of the model (7) would be a setting where also the population carrying labels 1 evolves over time, which is simulated in Section E.2. Most elements of the framework presented here would likely carry over the setting of three coupled PDEs: one for the evolution of $\rho(t)$, one for the evolution of $\bar{\rho}(t)$ and one for the classifier $\mu(t)$.

The term

$$\alpha \Delta \rho - \alpha \text{div}\left(\rho \nabla \log \tilde{\rho}\right) = \alpha \text{div}\left(\rho \nabla \delta_\rho KL(\rho \,|\, \tilde{\rho})\right)$$

forces the evolution of $\rho(t)$ to approach $\tilde{\rho}$. In other words, it penalizes (in energy) deviations from a given reference measure $\tilde{\rho}$. In the context of the application at hand, we take $\tilde{\rho}$ to be the initial distribution $\rho(t = 0)$. The solution $\rho(t)$ then evolves away from $\tilde{\rho}$ over time due to the other forces that are present. Therefore, the term $KL(\rho \,|\, \tilde{\rho})$ in the energy both provides smoothing of the flow and a penalization for deviations away from the reference measure $\tilde{\rho}$.

The self-interaction term $W * \rho$ introduces non-locality into the dynamics, as the decision for any given individual to move in a certain direction is influenced by the behavior of all other individuals in the population. The choice of $W$ is application dependent. Very often, the interaction between two individuals only depends on the distance between them. This suggests a choice of $W$ as a radial function, i.e. $W(z) = \omega(|z|)$. A choice of $\omega : \mathbb{R} \to \mathbb{R}$ such that $\omega'(r) > 0$ corresponds to an *attractive* force between individuals, whereas $\omega'(r) < 0$ corresponds to a *repulsive* force. The statement $|z|\omega'(|z|) = z \cdot \nabla_z W(z) \geq -D$ in Assumption 3 therefore corresponds to a requirement that the self-interaction force is not too repulsive. Neglecting all other forces in (7a), we obtain the non-local interaction equation

$$\partial_t \rho = \text{div}\left(\rho \nabla W * \rho\right)$$

which appears in many instances in mathematical biology, mathematical physics, and material science, and it is an equation that has been extensively studied over the past few decades, see for example [Car+11; BCY12; CCH14; BCL09; BLL12; CMV06; CFG23] and references therein. Using the results from these works, our assumptions on the interaction potential $W$ can be relaxed in many ways, for example by allowing discontinuous derivatives at zero for $W$, or by allowing $W$ to be negative.

The dynamics (7b) for the algorithm $\mu$ is a non-autonomous transport equation,

$$\partial_t \mu = \text{div}\left(\mu v\right),$$

where the time-dependence in the velocity field

$$v(t, x) := \nabla_x \left(\int f_1(z,x)\mathrm{d}\rho(t, z) + \int f_2(z,x)\mathrm{d}\tilde{\rho}(z) + \frac{\beta}{2}\|x - x_0\|^2\right),$$

comes through the evolving population $\rho(t)$. This structure allows to obtain an explicit solution for $\mu(t)$ in terms of the initial condition $\mu_0$ and the solution $\rho(t)$ to (7a) using the method of characteristics.

**Proposition 5.** *Assume that there exists a constant $c > 0$ such that*

$$\left\| \int \nabla_x f_1(z,x)\mathrm{d}\rho(z) + \int \nabla_x f_2(z,x)\mathrm{d}\bar\rho(z) \right\| \le c(1 + \|x\|) \quad \forall \rho \in \mathcal{P}_2(\mathbb{R}^d) \text{ and } \forall x \in \mathbb{R}^d. \tag{8}$$

*Then the unique distributional solution $\mu(t)$ to (7b) is given by*

$$\mu(t) = \Phi(t, 0, \cdot)_{\#}\mu_0, \tag{9}$$

*where $\Phi(t,s,x)$ solves the characteristic equation*

$$\partial_s \Phi(s,t,x) + v(s, \Phi(s,t,x)) = 0, \qquad \Phi(t,t,x) = x. \tag{10}$$

*Proof.* Thanks to Assumption 1, we have that $v \in C^1(\mathbb{R} \times \mathbb{R}^d; \mathbb{R}^d)$, and by (8), we have

$$\|v(t,x)\| \le c(1 + \|x\|) \quad \text{for all } t \ge 0, x \in \mathbb{R}^d.$$

By classical Cauchy-Lipschitz theory for ODEs, this guarantees the existence of a unique global solution $\Phi(t,s,x)$ solving (10). Then it can be checked directly that $\mu(t)$ as defined in (9) is a distributional solution to (7b). $\qquad\square$

In the characteristic equation (10), $\Phi(s,t,x)$ is a parametrization of all trajectories: if a particle was at location $x$ at time $t$, then it is at location $\Phi(s,t,x)$ at time $s$. Our assumptions on $f_1, f_2$ and $\bar\rho$ also ensure that $\Phi(s,t,\cdot) : \mathbb{R}^d \to \mathbb{R}^d$ is a $C^1$-diffeomorphism for all $s, t \in \mathbb{R}$. For more details on transport equations, see for example [De 07].

**Remark 2.** *Consider the special case where $\mu_0 = \delta_{x_0}$ for some initial position $x_0 \in \mathbb{R}^d$. Then by Proposition 5, the solution to (7b) is given by $\mu(t) = \delta_{x(t)}$, where $x(t) := \Phi(t, 0, x_0)$ solves the ODE*

$$\dot{x}(t) = -v(t, x(t)), \qquad x(0) = x_0,$$

*which is precisely of type (3).*

For the case of competing objectives, the two models we consider can be written as

$$\partial_t \rho = -\mathrm{div}\left(\rho\left[\nabla(f_1(z, b(\rho)) - \alpha \log(\rho/\tilde\rho) - W * \rho\right]\right),$$

$$b(\rho) := \underset{\bar x}{\mathrm{argmin}} \int f_1(z, \bar x)\mathrm{d}\rho(z) + \int f_2(\bar x, z')\mathrm{d}\bar\rho(z') + \frac{\beta}{2}\|\bar x - x_0\|^2$$

for (5), and

$$\frac{\mathrm{d}}{\mathrm{d}t}x = -\nabla_x\left(\int f_1(z, x)\,r(x)(\mathrm{d}z) + \int f_2(x, z')\mathrm{d}\bar\rho(z') + \frac{\beta}{2}\|x - x_0\|^2\right),$$

$$r(x) := \underset{\hat\rho \in \mathcal{P}}{\mathrm{argmax}} \int f_1(z, x)\mathrm{d}\hat\rho(z) - \alpha KL(\hat\rho|\tilde\rho) - \frac{1}{2}\int \hat\rho W * \hat\rho.$$

for (6).

## A.2 Definitions and notation

Here, and in what follows, $\mathrm{I}_d$ denotes the $d \times d$ identity matrix, and id denotes the identity map. The energy functionals we are considering are usually defined on the set of probability measures on $\mathbb{R}^d$, denoted by $\mathcal{P}(\mathbb{R}^d)$. If we consider the subset $\mathcal{P}_2(\mathbb{R}^d)$ of probability measures with bounded second moment,

$$\mathcal{P}_2(\mathbb{R}^d) := \left\{\rho \in \mathcal{P}(\mathbb{R}^d) : \int_{\mathbb{R}^d} \|z\|^2 \mathrm{d}\rho(z) < \infty\right\},$$

then we can endow this space with the Wasserstein-2 metric.

**Definition 4** (Wasserstein-2 Metric). *The Wasserstein-2 metric between two probability measures $\mu, \nu \in \mathcal{P}_2(\mathbb{R}^d)$ is given by*

$$W_2(\mu, \nu)^2 = \inf_{\gamma \in \Gamma(\mu, \nu)} \int \|z - z'\|_2^2 \, \mathrm{d}\gamma(z, z')$$

*where $\Gamma$ is the set of all joint probability distributions with marginals $\mu$ and $\nu$, i.e. $\mu(\mathrm{d}z) = \int \gamma(\mathrm{d}z, z')\mathrm{d}z'$ and $\nu(\mathrm{d}z') = \int \gamma(z, \mathrm{d}z')\mathrm{d}z$.*

The restriction to $\mathcal{P}_2(\mathbb{R}^d)$ ensures that $W_2$ is always finite. Then the space $(\mathcal{P}_2(\mathbb{R}^d), W_2)$ is indeed a metric space. We will make use of the fact that $W_2$ metrizes narrow convergence of probability measures. To make this statement precise, let us introduce two common notions of convergence for probability measures, which are a subset of the finite signed Radon measures $\mathcal{M}(\mathbb{R}^d)$.

**Definition 5.** *Consider a sequence $(\mu_n) \in \mathcal{M}(\mathbb{R}^d)$ and a limit $\mu \in \mathcal{M}(\mathbb{R}^d)$.*

- *(**Narrow topology**) The sequence $(\mu_n)$ converges* narrowly *to $\mu$, denoted by $\mu_n \rightharpoonup \mu$, if for all continuous bounded functions $f : \mathbb{R}^d \to \mathbb{R}$,*

$$\int_{\mathbb{R}^d} f(z) \mathrm{d}\mu_n(z) \to \int_{\mathbb{R}^d} f(z) \mathrm{d}\mu(z) \,.$$

- *(**Weak-$*$ topology**) The sequence $(\mu_n)$ converges* weakly-$*$ *to $\mu$, denoted by $\mu_n \xrightarrow{*} \mu$, if for all continuous functions vanishing at infinity (i.e. $f : \mathbb{R}^d \to \mathbb{R}$ such that for all $\epsilon > 0$ there exists a compact set $K_\epsilon \subset \mathbb{R}^d$ such that $|f(z)| < \epsilon$ on $\mathbb{R}^d \setminus K_\epsilon$), we have*

$$\int_{\mathbb{R}^d} f(z) \mathrm{d}\mu_n(z) \to \int_{\mathbb{R}^d} f(z) \mathrm{d}\mu(z) \,.$$

Let us denote the set of continuous functions on $\mathbb{R}^d$ vanishing at infinity by $C_0(\mathbb{R}^d)$, and the set of continuous bounded functions by $C_b(\mathbb{R}^d)$. Note that narrow convergence immediately implies that $\mu_n(\mathbb{R}^d) \to \mu(\mathbb{R}^d)$ as the constant function is in $C_b(\mathbb{R}^d)$. This is not necessarily true for weak-$*$ convergence. We will later make use of the Banach-Alaoglu theorem [Ala40], which gives weak-$*$ compactness of the unit ball in a dual space. Note that $\mathcal{M}(\mathbb{R}^d)$ is indeed the dual of $C_0(\mathbb{R}^d)$ endowed with the sup-norm, and $\mathcal{P}(\mathbb{R}^d)$ is the unit ball in $\mathcal{M}(\mathbb{R}^d)$ using the dual norm. Moreover, if we can ensure that mass does not escape to infinity, the two notions of convergence in Definition 5 are in fact equivalent.

**Lemma 6.** *Consider a sequence $(\mu_n) \in \mathcal{M}(\mathbb{R}^d)$ and a measure $\mu \in \mathcal{M}(\mathbb{R}^d)$. Then $\mu_n \rightharpoonup \mu$ if and only if $\mu_n \xrightarrow{*} \mu$ and $\mu_n(\mathbb{R}^d) \to \mu(\mathbb{R}^d)$.*

This follows directly from Definition 5. Here, the condition $\mu_n(\mathbb{R}^d) \to \mu(\mathbb{R}^d)$ is equivalent to tightness of $(\mu_n)$, and follows from Markov's inequality [Gho02] if we can establish uniform bounds on the second moments, i.e. we want to show that there exists a constant $C > 0$ independent of $n$ such that

$$\int \|z\|^2 \mathrm{d}\mu_n(z) < C \qquad \forall n \in \mathrm{N} \,. \tag{11}$$

**Definition 6** (Tightness of probability measures). *A collection of measures $(\mu_n) \in \mathcal{M}(\mathbb{R}^d)$ is* tight *if for all $\epsilon > 0$ there exists a compact set $K_\epsilon \subset \mathbb{R}^d$ such that $|\mu_n|(\mathbb{R}^d \setminus K_\epsilon) < \epsilon$ for all $n \in \mathrm{N}$, where $|\mu|$ denotes the total variation of $\mu$.*

Another classical result is that the Wasserstein-2 metric metrizes narrow convergence and weak-$*$ convergence of probability measures, see for example [San15, Theorem 5.11] or [Vil03, Theorem 7.12].

**Lemma 7.** *Let $\mu_n, \mu \in \mathcal{P}_2(\mathbb{R}^d)$. Then $W_2(\mu_n, \mu) \to 0$ if and only if*

$$\mu_n \rightharpoonup \mu \quad and \quad \int_{\mathbb{R}^d} \|z\|^2 \mathrm{d}\mu_n(z) \to \int_{\mathbb{R}^d} \|z\|^2 \mathrm{d}\mu(z) \,.$$

**Remark 3.** *Note that $\mu_n \rightharpoonup \mu$ can be replaced by $\mu_n \xrightarrow{*} \mu$ in the above statement thanks to the fact that the limit $\mu$ is a probability measure with mass 1, see Lemma 6.*

Next, we consider two measures $\mu, \nu \in \mathcal{P}(\mathbb{R}^d)$ that are *atomless*, i.e $\mu(\{z\}) = 0$ for all $z \in \mathbb{R}^d$. By Brenier's theorem [BB00] (also see [Vil03, Theorem 2.32]) there exists a unique measurable map $T : \mathbb{R}^d \to \mathbb{R}^d$ such that $T_\# \mu = \nu$, and $T = \nabla \psi$ for some convex function $\psi : \mathbb{R}^d \to \mathbb{R}$. Here, the *push-forward* operator $\nabla \psi_\#$ is defined as

$$\int_{\mathbb{R}^d} f(z) \mathrm{d}\nabla\psi_\# \rho_0(z) = \int_{\mathbb{R}^d} f(\nabla\psi(z)) \mathrm{d}\rho_0(z)$$

for all Borel-measurable functions $f : \mathbb{R}^d \mapsto \mathbb{R}_+$. If $\rho_1 = \nabla\psi_\# \rho_0$, we denote by $\rho_s = [(1-s)\,\mathrm{id} + s\nabla\psi]_\# \rho_0$ the *discplacement interpolant* between $\rho_0$ and $\rho_1$. We are now ready to introduce the notion of displacement convexity, which is the same as geodesic convexity in the geodesic space $(\mathcal{P}_2(\mathbb{R}^d), W_2)$. We will state the definition here for atomless measures, but it can be relaxed to any pair of measures in $\mathcal{P}_2$ using optimal transport plans instead of transport maps. In what follows, we will use $s$ to denote the interpolation parameter for geodesics, and $t$ to denote time related to solutions of (4), (5) and (6).

**Definition 7** (Displacement Convexity). *A functional $G : \mathcal{P} \mapsto \mathbb{R}$ is displacement convex if for all $\rho_0, \rho_1$ that are atomless we have*

$$G(\rho_s) \leq (1-s)G(\rho_0) + sG(\rho_1),$$

*where $\rho_s = [(1-s)\,\mathrm{id} + s\nabla\psi]_{\#}\rho_0$ is the displacement interpolant between $\rho_0$ and $\rho_1$. Further, $G : \mathcal{P} \mapsto \mathbb{R}$ is uniformly displacement convex with constant $\eta > 0$ if*

$$G(\rho_s) \leq (1-s)G(\rho_0) + sG(\rho_1) - s(1-s)\frac{\eta}{2}W_2(\rho_0, \rho_1)^2,$$

*where $\rho_s = [(1-s)\,\mathrm{id} + s\nabla\psi]_{\#}\rho_0$ is the displacement interpolant between $\rho_0$ and $\rho_1$.*

**Remark 4.** *In other words, $G$ is displacement convex (concave) if the function $G(\rho_s)$ is convex (concave) with $\rho_s = [(1 - s\,\mathrm{id} + s\nabla\psi]_{\#}\rho_0$ being the displacement interpolant between $\rho_0$ and $\rho_1$. Contrast this with the classical notion of convexity (concavity) for $G$, where we require that the function $G((1-s)\rho_0 + s\rho_1)$ is convex (concave).*

In fact, if the energy $G$ is twice differentiable along geodesics, then the condition $\frac{\mathrm{d}^2}{\mathrm{d}s^2}G(\gamma_s) \geq 0$ along any geodesic $(\rho_s)_{s \in [0,1]}$ between $\rho_0$ and $\rho_1$ is sufficient to obtain displacement convexity. Similarly, when $\frac{\mathrm{d}^2}{\mathrm{d}s^2}G(\rho_s) \geq \eta W_2(\rho_0, \rho_1)^2$, then $G$ is uniformly displacement convex with constant $\eta > 0$. For more details, see [McC97] and [Vil03, Chapter 5.2].

## A.3   Steady states

The main goal in our theoretical analysis is to characterize the asymptotic behavior for the models (4), (5) and (6) as time goes to infinity. The steady states of these equations are the natural candidates to be asymptotic profiles for the corresponding equations. Thanks to the gradient flow structure, we expect to be able to make a connection between ground states of the energy functionals, and the steady states of the corresponding gradient flow dynamics. More precisely, any minimizer or maximizer is in particular a critical point of the energy, and therefore satisfies that the first variation is constant on disconnected components of its support. If this ground state also has enough regularity (weak differentiability) to be a solution to the equation, it immediately follows that it is in fact a steady state.

To make this connection precise, we first introduce what exactly we mean by a steady state.

**Definition 8** (Steady states for (4)). *Given $\rho_\infty \in L^1_+(\mathbb{R}^d) \cap L^\infty_{loc}(\mathbb{R}^d)$ with $\|\rho_\infty\|_1 = 1$ and $\mu_\infty \in \mathcal{P}_2(\mathbb{R}^d)$, then $(\rho_\infty, \mu_\infty)$ is a steady state for the system (4) if $\rho_\infty \in W^{1,2}_{loc}(\mathbb{R}^d)$, $\nabla W * \rho_\infty \in L^1_{loc}(\mathbb{R}^d)$, $\rho_\infty$ is absolutely continuous with respect to $\tilde{\rho}$, and $(\rho_\infty, \mu_\infty)$ satisfy*

$$\nabla_z\left(\int f_1(z,x)\mathrm{d}\mu_\infty(x) + \alpha\log\left(\frac{\rho_\infty(z)}{\tilde{\rho}(z)}\right) + W * \rho_\infty(z)\right) = 0 \qquad \forall z \in \mathrm{supp}(\rho_\infty), \tag{12a}$$

$$\nabla_x\left(\int f_1(z,x)\mathrm{d}\rho_\infty(z) + \int f_2(z,x)\mathrm{d}\tilde{\rho}(z) + \frac{\beta}{2}\|x - x_0\|^2\right) = 0 \qquad \forall x \in \mathrm{supp}(\mu_\infty) \tag{12b}$$

*in the sense of distributions.*

Here, $L^1_+(\mathbb{R}^d) := \{\rho \in L^1(\mathbb{R}^d) : \rho \geq 0\}$.

**Definition 9** (Steady states for (5)). *Let $\rho_\infty \in L^1_+(\mathbb{R}^d) \cap L^\infty_{loc}(\mathbb{R}^d)$ with $\|\rho_\infty\|_1 = 1$. Then $\rho_\infty$ is a steady state for the system (5) if $\rho_\infty \in W^{1,2}_{loc}(\mathbb{R}^d)$, $\nabla W * \rho_\infty \in L^1_{loc}(\mathbb{R}^d)$, $\rho_\infty$ is absolutely continuous with respect to $\tilde{\rho}$, and $\rho_\infty$ satisfies*

$$\nabla_z\left(f_1(z, b(\rho_\infty)) - \alpha\log\left(\frac{\rho_\infty(z)}{\tilde{\rho}(z)}\right) - W * \rho_\infty(z)\right) = 0 \qquad \forall z \in \mathbb{R}^d, \tag{13}$$

*in the sense of distributions, where $b(\rho_\infty) := \mathrm{argmin}_x G_c(\rho_\infty, x)$.*

**Definition 10** (Steady states for (6)). *The vector $x_\infty \in \mathbb{R}^d$ is a steady state for the system (6) if it satisfies*

$$\nabla_x G_d(x_\infty) = 0.$$

In fact, with the above notions of steady state, we can obtain improved regularity for $\rho_\infty$.

**Lemma 8.** *Let Assumptions 1-3 hold. Then the steady states $\rho_\infty$ for (4) and (5) are continuous.*

*Proof.* We present here the argument for equation (5) only. The result for (4) follows in exactly the same way by replacing $f_1(z, b(\rho_\infty))$ with $-\int f_1(z,x)\mathrm{d}\mu_\infty(x)$.

Thanks to our assumptions, we have $f_1(\cdot, b(\rho_\infty)) + \alpha \log \tilde{\rho}(\cdot) \in C^1$, which implies that $\nabla(f_1(\cdot, b(\rho_\infty)) + \alpha \log \tilde{\rho}(\cdot)) \in L^\infty_{loc}$. By the definition of a steady state, $\rho_\infty \in L^1 \cap L^\infty_{loc}$ and thanks to Assumption 3 we have $W \in C^2$, which implies that $\nabla W * \rho_\infty \in L^\infty_{loc}$. Let

$$h(z) := \rho_\infty(z) \nabla \left[ f_1(z, b(\rho_\infty)) + \alpha \log \tilde{\rho}(z) - (W * \rho_\infty)(z) \right].$$

Then by the aforementioned regularity, we obtain $h \in L^1_{loc} \cap L^\infty_{loc}$. By interpolation, it follows that $h \in L^p_{loc}$ for all $1 < p < \infty$. This implies that $\text{div}(h) \in W^{-1,p}_{loc}$. Since $\rho_\infty$ is a weak $W^{1,2}_{loc}$-solution of (13), we have

$$\Delta \rho_\infty = \text{div}(h),$$

and so by classic elliptic regularity theory we conclude $\rho_\infty \in W^{1,p}_{loc}$. Finally, applying Morrey's inequality, we have $\rho_\infty \in C^{0,k}$ where $k = \frac{p-d}{p}$ for any $d < p < \infty$. Therefore $\rho_\infty \in C(\mathbb{R}^d)$ (after possibly being redefined on a set of measure zero). $\qquad \square$

# B    Proof of Theorem 2

For ease of notation, we write $G_a : \mathcal{P}(\mathbb{R}^d) \times \mathcal{P}(\mathbb{R}^d) \mapsto [0, \infty]$ as

$$G_a((\rho, \mu)) = \alpha KL(\rho|\tilde{\rho}) + \mathcal{V}(\rho, \mu) + \mathcal{W}(\rho),$$

where we define

$$\mathcal{V}(\rho, \mu) = \iint f_1(z, x) \mathrm{d}\rho(z) \mathrm{d}\mu(x) + \int V(x) \mathrm{d}\mu(x),$$

$$\mathcal{W}(\rho) = \frac{1}{2} \iint W(z_1 - z_2) \mathrm{d}\rho(z_1) \mathrm{d}\rho(z_2),$$

with potential given by $V(x) := \int f_2(z, x) \mathrm{d}\bar{\rho}(z) + \frac{\beta}{2} \|x - x_0\|^2$.

In order to prove the existence of a unique ground state for $G_a$, a natural approach is to consider the corresponding Euler-Lagrange equations

$$\alpha \log \frac{\rho(z)}{\tilde{\rho}(z)} + \int f_1(z, x) \mathrm{d}\mu(x) + (W * \rho)(z) = c_1[\rho, \mu] \quad \text{for all } z \in \text{supp}(\rho), \tag{14a}$$

$$\int f_1(z, x) \mathrm{d}\rho(z) + V(x) = c_2[\rho, \mu] \quad \text{for all } x \in \text{supp}(\mu), \tag{14b}$$

where $c_1, c_2$ are constants that may differ on different connected components of $\text{supp}(\rho)$ and $\text{supp}(\mu)$. These equations are not easy to solve explicitly, and we are therefore using general non-constructive techniques from calculus of variations. We first show continuity and convexity properties for the functional $G_a$ (Lemma 9 and Proposition 10), essential properties that will allow us to deduce existence and uniqueness of ground states using the direct method in the calculus of variations (Proposition 11). Using the Euler-Lagrange equation 14, we then prove properties on the support of the ground state (Corollary 12). To obtain convergence results, we apply the HWI method: we first show a general 'interpolation' inequality between the energy, the energy dissipation and the metric (Proposition 13); this fundamental inequality will then imply a generalized logarithmic Sobolev inequality (Corollary 14) relating the energy to the energy dissipation, and a generalized Talagrand inequality (Corollary 15) that allows to translate convergence in energy into convergence in metric. Putting all these ingrediends together will then allow us to conclude for the statements in Theorem 2.

**Lemma 9** (Lower semi-continuity)**.** *Let Assumptions 1-3 hold. Then the functional $G_a : \mathcal{P} \times \mathcal{P} \to \mathbb{R}$ is lower semi-continuous with respect to the weak-$*$ topology.*

*Proof.* We split the energy $G_a$ into three parts: (i) $KL(\rho|\tilde{\rho})$, (ii) the interaction energy $\mathcal{W}$, and (iii) the potential energy $\mathcal{V}$. For (i), lower semi-continuity has been shown in [Pos75]. For (ii), we can directly apply [San15, Proposition 7.2] using Assumption 3. For (iii), note that $V$ and $f_1$ are lower semi-continuous and bounded below thanks to Assumption 1, and so the result follows from [San15, Proposition 7.1]. $\qquad \square$

**Proposition 10** (Uniform displacement convexity)**.** *Let $\alpha, \beta > 0$. Fix $\gamma_0, \gamma_1 \in \mathcal{P}_2 \times \mathcal{P}_2$ and let Assumptions 1-3 hold. Along any geodesic $(\gamma_s)_{s \in [0,1]} \in \mathcal{P}_2 \times \mathcal{P}_2$ connecting $\gamma_0$ to $\gamma_1$, we have for all $s \in [0, 1]$*

$$\frac{\mathrm{d}^2}{\mathrm{d}s^2} G_a(\gamma_s) \geq \lambda_a \overline{W}(\gamma_0, \gamma_1)^2, \qquad \lambda_a := \lambda_1 + \min(\lambda_2 + \beta, \alpha \tilde{\lambda}). \tag{15}$$

*As a result, the functional $G_a : \mathcal{P} \times \mathcal{P} \to \mathbb{R}$ is uniformly displacement convex with constant $\lambda_a > 0$.*

*Proof.* Let $\gamma_0$ and $\gamma_1$ be two probability measures with bounded second moments. Denote by $\phi, \psi : \mathbb{R}^d \to \mathbb{R}$ the optimal Kantorovich potentials pushing $\rho_0$ onto $\rho_1$, and $\mu_0$ onto $\mu_1$, respectively:

$$\rho_1 = \nabla\phi_\# \rho_0 \quad \text{such that} \quad W_2(\rho_0, \rho_1)^2 = \int_{\mathbb{R}^d} \|z - \nabla\phi(z)\|^2 \mathrm{d}\rho_0(z)\,,$$

$$\mu_1 = \nabla\psi_\# \mu_0 \quad \text{such that} \quad W_2(\mu_0, \mu_1)^2 = \int_{\mathbb{R}^d} \|x - \nabla\psi(x)\|^2 \mathrm{d}\mu_0(x)\,.$$

The now classical results in [BB00] guarantee that there always exists convex functions $\phi, \psi$ that satisfy the conditions above. Then the path $(\gamma_s)_{s \in [0,1]} = (\rho_s, \mu_s)_{s \in [0,1]}$ defined by

$$\rho_s = [(1-s)\,\mathrm{id} + s\nabla_z\phi]_\# \rho_0\,,$$
$$\mu_s = [(1-s)\,\mathrm{id} + s\nabla_x\psi]_\# \mu_0$$

is a $\overline{W}$-geodesic from $\gamma_0$ to $\gamma_1$.

The first derivative of $\mathcal{V}$ along geodesics in the Wasserstein metric is given by

$$
\begin{aligned}
\frac{\mathrm{d}}{\mathrm{d}s}\mathcal{V}(\gamma_s) = & \frac{\mathrm{d}}{\mathrm{d}s} \Bigg[ \iint f_1((1-s)z + s\nabla\phi(z), (1-s)x + s\nabla\psi(x))\, \mathrm{d}\rho_0(z)\mathrm{d}\mu_0(x) \\
& + \int V((1-s)x + s\nabla\psi(x))\, \mathrm{d}\mu_0(x) \Bigg] \\
= & \iint \nabla_x f_1((1-s)z + s\nabla\phi(z), (1-s)x + s\nabla\psi(x)) \cdot (\nabla\psi(x) - x)\, \mathrm{d}\rho_0(z)\mathrm{d}\mu_0(x) \\
& \iint \nabla_z f_1((1-s)z + s\nabla\phi(z), (1-s)x + s\nabla\psi(x)) \cdot (\nabla\phi(z) - z)\, \mathrm{d}\rho_0(z)\mathrm{d}\mu_0(x) \\
& + \int \nabla_x V((1-s)x + s\nabla\psi(x)) \cdot (\nabla\psi(x) - x)\, \mathrm{d}\mu_0(x)\,,
\end{aligned}
$$

and taking another derivative we have

$$
\begin{aligned}
\frac{\mathrm{d}^2}{\mathrm{d}s^2}\mathcal{V}(\gamma_s) = & -\iint \begin{bmatrix} (\nabla\psi(x) - x) \\ (\nabla\phi(z) - z) \end{bmatrix}^T \cdot D_s(z,x) \cdot \begin{bmatrix} (\nabla\psi(x) - x) \\ (\nabla\phi(z) - z) \end{bmatrix} \mathrm{d}\rho_0(z)\mathrm{d}\mu_0(x) \\
& + \iint (\nabla\psi(x) - x)^T \cdot \nabla_x^2 V((1-s)x + s\nabla\psi(x)) \cdot (\nabla\psi(x) - x)\, \mathrm{d}\rho_0(z)\mathrm{d}\mu_0(x) \\
& \geq \lambda_1 \overline{W}(\gamma_0, \gamma_1)^2 + (\lambda_2 + \beta) W_2(\mu_0, \mu_1)^2\,,
\end{aligned}
$$

where we denoted $D_s(z,x) := \mathrm{Hess}(f_1)((1-s)z + s\nabla\phi(z), (1-s)x + s\nabla\psi(x))$, and the last inequality follows from Assumption 1 and the optimality of the potentials $\phi$ and $\psi$.

Following [CMV03; Vil03] and using Assumption 2, the second derivatives of the diffusion term and the interaction term along geodesics are given by

$$\frac{\mathrm{d}^2}{\mathrm{d}s^2} KL(\rho_s | \tilde\rho) \geq \alpha\tilde\lambda\, W_2(\rho_0, \rho_1)^2\,, \qquad \frac{\mathrm{d}^2}{\mathrm{d}s^2}\mathcal{W}(\rho_s) \geq 0. \tag{16}$$

Putting the above estimates together, we obtain (15).

$\square$

**Remark 5.** *Alternatively, one could assume strong convexity of $W$, which would improve the lower-bound on the second derivative along geodesics.*

**Proposition 11.** *(Ground state) Let Assumptions 1-3 hold for $\alpha, \beta > 0$. Then the functional $G_a : \mathcal{P}(\mathbb{R}^d) \times \mathcal{P}(\mathbb{R}^d) \to [0, \infty]$ admits a unique minimizer $\gamma_* = (\rho_*, \mu_*)$, and it satisfies $\rho_* \in \mathcal{P}_2(\mathbb{R}^d) \cap L^1(\mathbb{R}^d)$, $\mu_* \in \mathcal{P}_2(\mathbb{R}^d)$, and $\rho_*$ is absolutely continuous with respect to $\tilde\rho$.*

*Proof.* We show existence of a minimizer of $G_a$ using the direct method in the calculus of variations. Denote by $\gamma = (\rho, \mu) \in \mathcal{P} \times \mathcal{P} \subset \mathcal{M} \times \mathcal{M}$ a pair of probability measures as a point in the product space of Radon measures. Since $G_a \geq 0$ on $\mathcal{P} \times \mathcal{P}$ (see Assumption 1) and not identically $+\infty$ everywhere, there exists a minimizing sequence $(\gamma_n) \in \mathcal{P} \times \mathcal{P}$. Note that $(\gamma_n)$ is in the closed unit ball of the dual space of continuous functions vanishing at infinity $(C_0(\mathbb{R}^d) \times C_0(\mathbb{R}^d))^*$ endowed with the dual norm $\|\gamma_n\|_* = \sup \frac{|\int f\mathrm{d}\rho_n + \int g\mathrm{d}\mu_n|}{\|(f,g)\|_\infty}$ over $f, g \in C_0(\mathbb{R}^d)$ with $\|(f,g)\|_\infty := \|f\|_\infty + \|g\|_\infty \neq 0$. By the Banach-Alaoglu theorem [Rud91, Thm 3.15] there exists a limit $\gamma_* = (\rho_*, \mu_*) \in \mathcal{M} \times \mathcal{M} = (C_0 \times C_0)^*$ and a convergent subsequence (not relabelled) such that $\gamma_n \overset{*}{\rightharpoonup} \gamma_*$. In fact, since $KL(\rho_* | \tilde\rho) < \infty$ it follows that $\rho_*$ is absolutely continuous with respect to $\tilde\rho$, implying $\rho_* \in L^1(\mathbb{R}^d)$ thanks to Assumption 2. Further, $\mu_*$ has bounded second moment, else we would have

$\inf_{\gamma \in \mathcal{P} \times \mathcal{P}} G_a(\gamma) = \infty$ which yields a contradiction. It remains to show that $\int \mathrm{d}\rho_* = \int \mathrm{d}\mu_* = 1$ to conclude that $\gamma_* \in \mathcal{P} \times \mathcal{P}$. To this aim, it is sufficient to show tightness of $(\rho_n)$ and $(\mu_n)$, preventing the escape of mass to infinity as we have $\int \mathrm{d}\rho_n = \int \mathrm{d}\mu_n = 1$ for all $n \geq 1$. Tightness follows from Markov's inequality [Gho02] if we can establish uniform bounds on the second moments, i.e. we want to show that there exists a constant $C > 0$ independent of $n$ such that

$$\int \|z\|^2 \mathrm{d}\rho_n(z) + \int \|x\|^2 \mathrm{d}\mu_n(x) < C \qquad \forall n \in \mathrm{N}. \tag{17}$$

To establish (17), observe that thanks to Assumption 2, there exists a constant $c_0 \in \mathbb{R}$ (possibly negative) such that $-\log \tilde{\rho}(z) \geq c_0 + \frac{\tilde{\lambda}}{4}\|z\|^2$ for all $z \in \mathbb{R}^d$. Then

$$\frac{\alpha \tilde{\lambda}}{4} \int \|z\|^2 \mathrm{d}\rho_n \leq -\alpha c_0 - \alpha \int \log \tilde{\rho}(z) \mathrm{d}\rho_n$$

Therefore, using $\int \mathrm{d}\rho_n = \int \mathrm{d}\mu_n = 1$ and writing $\zeta := \min\{\frac{\alpha \tilde{\lambda}}{4}, \frac{\beta}{2}\} > 0$, we obtain the desired uniform upper bound on the second moments of the minimizing sequence,

$$\zeta \iint \left(\|z\|^2 + \|x\|^2\right) \mathrm{d}\rho_n \mathrm{d}\mu_n \leq -\alpha c_0 - \alpha \int \log \tilde{\rho}(z) \mathrm{d}\rho_n + \beta \int \|x - x_0\|^2 \mathrm{d}\mu_n + \beta \|x_0\|^2$$

$$\leq -\alpha c_0 + \beta \|x_0\|^2 + G_a(\gamma_n)$$

$$\leq -\alpha c_0 + \beta \|x_0\|^2 + G_a(\gamma_1) < \infty.$$

This concludes the proof that the limit $\gamma_*$ satisfies indeed $\gamma_* \in \mathcal{P} \times \mathcal{P}$, and indeed $\rho_* \in \mathcal{P}_2(\mathbb{R}^d)$ as well. Finally, $\gamma_*$ is indeed a minimizer of $G_a$ thanks to weak-* lower-semicontinuity of $G_a$ following Lemma 9.

Next we show uniqueness using a contradiction argument. Suppose $\gamma_* = (\rho_*, \mu_*)$ and $\gamma'_* = (\rho'_*, \mu'_*)$ are minimizers of $G_a$. For $s \in [0,1]$, define $\gamma_s := ((1-s)\,\mathrm{id} + sT, (1-s)\,\mathrm{id} + sS)_{\#}\gamma_*$, where $T, S : \mathbb{R}^d \mapsto \mathbb{R}^d$ are the optimal transport maps such that $\rho'_* = T_{\#}\rho_*$ and $\mu'_* = S_{\#}\mu_*$. By Proposition 10 the energy $G_a$ is uniformly displacement convex, and so we have

$$G_a(\gamma_s) \leq (1-s)G_a(\gamma_*) + sG_a(\gamma'_*) = G_a(\gamma_*).$$

If $\gamma_* \neq \gamma'_*$ and $s \in (0,1)$, then strict inequality holds by applying similar arguments as in [McC97, Proposition 1.2]. However, the strict inequality $G_a(\gamma_s) < G_a(\gamma_*)$ for $\gamma_* \neq \gamma'_*$ is a contradiction to the minimality of $\gamma_*$. Hence, the minimizer is unique. $\qquad \square$

**Remark 6.** *If $\lambda_1 > 0$, then the strict convexity of $f_1$ can be used to deduce uniqueness, and the assumptions on $-\log \tilde{\rho}$ can be weakened from strict convexity to convexity.*

**Corollary 12.** *Let Assumptions 1-3 hold. Any minimizer $\gamma_* = (\rho_*, \mu_*)$ of $G_a$ is a steady state for equation (4) according to Definition 8 and satisfies $\mathrm{supp}(\rho_*) = \mathrm{supp}(\tilde{\rho})$.*

*Proof.* By Proposition 11, we have $\rho_*, \mu_* \in \mathcal{P}_2$, as well as $\rho_* \in L^1_+$, $\|\rho_*\|_1 = 1$, and that $\rho_*$ is absolutely continuous with respect to $\tilde{\rho}$. Since $W \in C^2(\mathbb{R}^d)$, it follows that $\nabla W * \rho_* \in L^1_{loc}$. In order to show that $\gamma_*$ is a steady state for equation (4), it remains to prove that $\rho_* \in W^{1,2}_{loc} \cap L^\infty_{loc}$. As $\gamma^*$ is a minimizer, it is in particular a critical point, and therefore satisfies equations (14). Rearranging, we obtain (for a possible different constant $c_1[\rho_*, \mu_*] \neq 0$) from (14a) that

$$\rho_*(z) = c_1[\rho_*, \mu_*]\tilde{\rho}(z) \exp\left[-\frac{1}{\alpha}\left(\int f_1(z, x)\,\mu_*(x) + W * \rho_*(z)\right)\right] \qquad \text{on } \mathrm{supp}(\rho_*). \tag{18}$$

Then for any compact set $K \subset \mathbb{R}^d$,

$$\sup_{z \in K} \rho_*(z) \leq c_1[\rho_*, \mu_*] \sup_{z \in K} \tilde{\rho}(z) \sup_{z \in K} \exp\left(-\frac{1}{\alpha}\left(\int f_1(z, x)\,\mu_*(x)\right)\right) \sup_{z \in K} \exp\left(-\frac{1}{\alpha}W * \rho_*\right).$$

As $f_1 \geq 0$ by Assumption 1 and $W \geq 0$ by Assumption 3, the last two terms are finite. The first supremum is finite thanks to continuity of $\tilde{\rho}$. Therefore $\rho_* \in L^\infty_{loc}$. To show that $\rho_* \in W^{1,2}_{loc}$, note that for any compact set $K \subset \mathbb{R}^d$, we have $\int_K |\rho_*(z)|^2 \mathrm{d}z < \infty$ as a consequence of $\rho_* \in L^\infty_{loc}$. Moreover, defining $T[\gamma](z) := -\frac{1}{\alpha}\left(\int f_1(z, x)\,\mu(x) + W * \rho(z)\right) \leq 0$, we have

$$\int_K |\nabla \rho_*|^2 \mathrm{d}z = c_1[\rho_*, \mu_*]^2 \int_K |\nabla \tilde{\rho} + \tilde{\rho}\nabla T[\gamma_*]|^2 \exp(2T[\gamma_*]) \mathrm{d}z$$

$$\leq 2c_1[\rho_*, \mu_*]^2 \int_K |\nabla \tilde{\rho}|^2 \exp(2T[\gamma_*]) \mathrm{d}z + 2c_1[\rho_*, \mu_*]^2 \int_K |\nabla T[\gamma_*]|^2 \tilde{\rho}^2 \exp(2T[\gamma_*]) \mathrm{d}z,$$

which is bounded noting that $\exp(2T[\gamma_*]) \leq 1$ and that $T[\gamma_*](\cdot), \nabla T[\gamma_*](\cdot)$ and $\nabla \tilde{\rho}$ are in $L^\infty_{loc}$, where we used that $f_1, (\cdot, x), W(\cdot), \tilde{\rho}(\cdot) \in C^1(\mathbb{R}^d)$ by Assumptions 1-3. We conclude that $\rho_* \in W^{1,2}_{loc}$, and indeed $(\rho_*, \mu_*)$ solves (12) in the sense of distributions as a consequence of (14).

Next, we show that $\mathrm{supp}(\rho_*) = \mathrm{supp}(\tilde{\rho})$ using again the relation (18). Firstly, note that $\mathrm{supp}(\rho_*) \subset \mathrm{supp}(\tilde{\rho})$ since $\rho_*$ is absolutely continuous with respect to $\tilde{\rho}$. Secondly, we claim that $\exp\left[-\frac{1}{\alpha}\left(\int f_1(z,x)\,\mu_*(x) + W * \rho_*(z)\right)\right] > 0$ for all $z \in \mathbb{R}^d$. In other words, we claim that $\int f_1(z,x)\,\mu_*(x) < \infty$ and $W * \rho_*(z) < \infty$ for all $z \in \mathbb{R}^d$. Indeed, for the first term, fix any $z \in \mathbb{R}^d$ and choose $R > 0$ large enough such that $z \in B_R(0)$. Then, thanks to continuity of $f_1$ according to Assumption 1, we have

$$\int f_1(z,x)\,\mu_*(x) \leq \sup_{z \in B_R(0)} \int f_1(z,x)\,\mu_*(x) < \infty\,.$$

For the second term, note that by Assumption 3, we have for any $z \in \mathbb{R}^d$ and $\epsilon > 0$,

$$W(z) \leq W(0) + \nabla W(z) \cdot z \leq W(0) + \frac{1}{2\epsilon}\|\nabla W(z)\|^2 + \frac{\epsilon}{2}\|z\|^2$$

$$\leq W(0) + \frac{D^2}{2\epsilon}(1 + \|z\|)^2 + \frac{\epsilon}{2}\|z\|^2 \leq W(0) + \frac{D^2}{\epsilon} + \left(\frac{D^2}{\epsilon} + \frac{\epsilon}{2}\right)\|z\|^2$$

$$= W(0) + \frac{D}{\sqrt{2}} + \sqrt{2}D\|z\|^2\,,$$

where the last equality follows by choosing the optimal $\epsilon = \sqrt{2}D$. We conclude that

$$W * \rho_*(z) \leq W(0) + \frac{D}{\sqrt{2}} + \sqrt{2}D \int \|z - \tilde{z}\|^2\,\rho_*(\tilde{z})$$

$$\leq W(0) + \frac{D}{\sqrt{2}} + 2\sqrt{2}D\|z\|^2 + 2\sqrt{2}D \int \|\tilde{z}\|^2\,\rho_*(\tilde{z})\,, \tag{19}$$

which is finite for any fixed $z \in \mathbb{R}^d$ thanks to the fact that $\rho_* \in \mathcal{P}_2(\mathbb{R}^d)$. Hence, $\mathrm{supp}(\rho_*) = \mathrm{supp}(\tilde{\rho})$. $\qquad \square$

**Remark 7.** *If we have in addition that $\tilde{\rho} \in L^\infty(\mathbb{R}^d)$, then the minimizer $\rho_*$ of $G_a$ is in $L^\infty(\mathbb{R}^d)$ as well. This follows directly by bounding the right-hand side of* (18).

The following inequality is referred to as HWI inequality and represents the key result to obtain convergence to equilibrium.

**Proposition 13** (HWI inequality)**.** *Define the dissipation functional*

$$D_a(\gamma) := \iint |\nabla_{x,z}\delta_\gamma G_a(z,x)|^2 \mathrm{d}\gamma(z,x)\,.$$

*Assume $\alpha, \beta > 0$ and let $\lambda_a$ as defined in* (15)*. Let $\gamma_0, \gamma_1 \in \mathcal{P}_2 \times \mathcal{P}_2$ such that $G_a(\gamma_0), G_a(\gamma_1), D_a(\gamma_0) < \infty$. Then*

$$G_a(\gamma_0) - G_a(\gamma_1) \leq \overline{W}(\gamma_0, \gamma_1)\sqrt{D_a(\gamma_0)} - \frac{\lambda_a}{2}\overline{W}(\gamma_0, \gamma_1)^2 \tag{20}$$

*Proof.* For simplicity, consider $\gamma_0, \gamma_1$ that have smooth Lebesgue densities of compact support. The general case can be recovered using approximation arguments. Let $(\gamma_s)_{s \in [0,1]}$ denote a $\overline{W}$-geodesic between $\gamma_0, \gamma_1$. Following similar arguments as in [CMV03] and [OV00, Section 5] and making use of the calculations in the proof of Proposition 10, we have

$$\frac{\mathrm{d}}{\mathrm{d}s}G_a(\gamma_s)\bigg|_{s=0} \geq \iint \begin{bmatrix} \xi_1(z) \\ \xi_2(x) \end{bmatrix} \cdot \begin{bmatrix} (\nabla\phi(z) - z) \\ (\nabla\psi(x) - x) \end{bmatrix} \mathrm{d}\gamma_0(z,x)\,,$$

where

$$\xi_1[\gamma_0](z) := \alpha\nabla_z \log\left(\frac{\rho_0(z)}{\tilde{\rho}(z)}\right) + \int \nabla_z f_1(z,x)\mathrm{d}\mu_0(x) + \int \nabla_z W(z - z')\mathrm{d}\rho_0(z')\,,$$

$$\xi_2[\gamma_0](x) := \int \nabla_x f_1(z,x)\mathrm{d}\rho_0(z) + \nabla_x V(x)\,.$$

Note that the dissipation functional can then be written as

$$D_a(\gamma_0) = \iint \left(|\xi_1(z)|^2 + |\xi_2(x)|^2\right) \mathrm{d}\gamma_0(z,x)\,.$$

Using the double integral Cauchy-Schwarz inequality [Ste04], we obtain

$$\frac{\mathrm{d}}{\mathrm{d}s}G_a(\gamma_s)\Big|_{s=0} \geq -\left(\sqrt{\iint \left\|\begin{bmatrix}\xi_1\\\xi_2\end{bmatrix}\right\|_2^2 \mathrm{d}\gamma_0}\right)\left(\sqrt{\iint \left\|\begin{bmatrix}\nabla\phi(z)-z\\\nabla\psi(x)-x\end{bmatrix}\right\|_2^2 \mathrm{d}\gamma_0}\right)$$

$$= -\sqrt{D_a(\gamma_0)}\sqrt{\int \|\nabla\phi(z)-z\|^2 \mathrm{d}\rho_0 + \int \|\nabla\psi(x)-x\|^2 \mathrm{d}\mu_0}$$

$$= -\sqrt{D_a(\gamma_0)}\,\overline{W}(\gamma_0,\gamma_1)\,.$$

Next, we compute a Taylor expansion of $G_a(\gamma_s)$ when considered as a function in $s$ and use the bound on $\frac{\mathrm{d}^2}{\mathrm{d}s^2}G_a$ from (15):

$$G_a(\gamma_1) = G_a(\gamma_0) + \frac{\mathrm{d}}{\mathrm{d}s}G_a(\gamma_s)\Big|_{s=0} + \int_0^1 (1-t)\left(\frac{\mathrm{d}^2}{\mathrm{d}s^2}G_a(\gamma_s)\right)\Big|_{s=t}\mathrm{d}t$$

$$\geq G_a(\gamma_0) - \sqrt{D_a(\gamma_0)}\,\overline{W}(\gamma_0,\gamma_1) + \frac{\lambda_a}{2}\overline{W}(\gamma_0,\gamma_1)^2\,.$$

$\square$

**Remark 8.** *The HWI inequality in Proposition 13 immediately implies uniqueness of minimizers for $G_a$ in the set $\{\gamma \in \mathcal{P} \times \mathcal{P} : D_a(\gamma) < +\infty\}$. Indeed, if $\gamma_0$ is such that $D_a(\gamma_0) = 0$, then for any other $\gamma_1$ in the above set we have $G_a(\gamma_0) \leq G_a(\gamma_1)$ with equality if and only if $\overline{W}(\gamma_0,\gamma_1) = 0$.*

**Corollary 14** (Generalized Log-Sobolev inequality). *Denote by $\gamma_*$ the unique minimizer of $G_a$. With $\lambda_a$ as defined in (15), any product measure $\gamma \in \mathcal{P}_2 \times \mathcal{P}_2$ such that $G(\gamma), D_a(\gamma) < \infty$ satisfies*

$$D_a(\gamma) \geq 2\lambda_a\, G_a(\gamma|\gamma_*)\,. \tag{21}$$

*Proof.* This statement follows immediately from Proposition 13. Indeed, let $\gamma_1 = \gamma_*$ and $\gamma_0 = \gamma$ in (20). Then

$$G_a(\gamma \mid \gamma_*) \leq \overline{W}(\gamma,\gamma_*)\sqrt{D_a(\gamma)} - \frac{\lambda_a}{2}\overline{W}(\gamma,\gamma_*)^2$$

$$\leq \max_{t\geq 0}\left(\sqrt{D_a(\gamma)}t - \frac{\lambda_a}{2}t^2\right) = \frac{D_a(\gamma)}{2\lambda_a}\,.$$

$\square$

**Corollary 15** (Talagrand inequality). *Denote by $\gamma_*$ the unique minimizer of $G_a$. With $\lambda_a$ as defined in (15), it holds*

$$\overline{W}(\gamma,\gamma_*)^2 \leq \frac{2}{\lambda_a}G_a(\gamma \mid \gamma_*)$$

*for any $\gamma \in \mathcal{P}_2 \times \mathcal{P}_2$ such that $G_a(\gamma) < \infty$.*

*Proof.* This is also a direct consequence of Proposition 13 by setting $\gamma_0 = \gamma_*$ and $\gamma_1 = \gamma$. Then $G_a(\gamma_*) < \infty$ and $D_a(\gamma_*) = 0$, and the result follows. $\square$

*Proof of Theorem 2.* The entropy term $\int \rho\log\rho$ produces diffusion in $\rho$ for the corresponding PDE in (4). As a consequence, solutions $\rho_t$ to (4) and minimizers $\rho^*$ for $G_a$ have to be $L^1$ functions. As there is no diffusion for the evolution of $\mu_t$, solutions may have a singular part. In fact, for initial condition $\mu_0 = \delta_{x_0}$, the corresponding solution will be of the form $\mu_t = \delta_{x(t)}$, where $x(t)$ solves the ODE (3) with initial condition $x_0$. This follows from the fact that the evolution for $\mu_t$ is a transport equation (also see Section A.1 for more details). Results (a) and (b) are the statements in Proposition 11, Corollary 12 and Corollary 15. To obtain (c), we differentiate the energy $G_a$ along solutions $\gamma_t$ to the equation (4):

$$\frac{\mathrm{d}}{\mathrm{d}t}G_a(\gamma_t) = \int \delta_\rho G_a[\gamma_t](z)\partial_t\rho_t\mathrm{d}z + \int \delta_\mu G_a[\gamma_t](x)\partial_t\mu_t\mathrm{d}x$$

$$= -\int \|\nabla_z\delta_\rho G_a[\gamma_t](z)\|^2\,\mathrm{d}\rho_t(z) - \int \|\nabla_x\delta_\mu G_a[\gamma_t](x)\|^2\,\mathrm{d}\mu_t(x)$$

$$= -D_a(\gamma_t) \leq -2\lambda_a G_a(\gamma_t \mid \gamma_*)\,,$$

where the last bound follows from Corollary 14. Applying Gronwall's inequality, we immediately obtain decay in energy,

$$G_a(\gamma_t \mid \gamma_*) \leq e^{-2\lambda_a t}G_a(\gamma_0 \mid \gamma_*)\,.$$

Finally, applying Talagrand's inequality (Corollary 15), the decay in energy implies decay in the product Wasserstein metric,

$$\overline{W}(\gamma_t,\gamma_*) \leq ce^{-\lambda_a t}$$

where $c > 0$ is a constant only depending on $\gamma_0$, $\gamma_*$ and the parameter $\lambda_a$. $\square$

# C  Proof of Theorem 3

In the case of competing objectives, we rewrite the energy $G_c(\rho, x) : \mathcal{P}(\mathbb{R}^d) \times \mathbb{R}^d \mapsto [-\infty, \infty]$ as follows:

$$G_c(\rho, x) = \int f_1(z, x) \mathrm{d}\rho(z) + \int f_2(z, x) \mathrm{d}\bar{\rho}(z) + \frac{\beta}{2} \|x - x_0\|^2 - P(\rho) \,,$$

where

$$P(\rho) := \alpha KL(\rho|\tilde{\rho}) + \frac{1}{2} \int \rho W * \rho \,.$$

Note that for any fixed $\rho \in \mathcal{P}$, the energy $G_c(\rho, \cdot)$ is strictly convex in $x$, and therefore has a unique minimizer. Define the best response by

$$b(\rho) := \operatorname*{argmin}_{\bar{x}} G_c(\rho, \bar{x})$$

and denote $G_b(\rho) := G_c(\rho, b(\rho))$. We begin with auxiliary results computing the first variations of the best response $b$ and then the different terms in $G_b(\rho)$ using Definition 1.

**Lemma 16** (First variation of the best response). *The first variation of the best response of the classifier at $\rho$ (if it exists) is*

$$\delta_\rho b[\rho](z) = -Q(\rho)^{-1} \nabla_x f_1(z, b(\rho)) \quad \text{for almost every } z \in \mathbb{R}^d \,,$$

*where $Q(\rho) \succeq (\beta + \lambda_1 + \lambda_2) \, \mathrm{I}_d$ is a symmetric matrix, constant in $z$ and $x$, defined as*

$$Q(\rho) := \beta \, \mathrm{I}_d + \int \nabla_x^2 f_1(z, b(\rho)) \mathrm{d}\rho(z) + \int \nabla_x^2 f_2(z, b(\rho)) \mathrm{d}\bar{\rho}(z) \,.$$

*In particular, we then have for any $\psi \in C_c^\infty(\mathbb{R}^d)$ with $\int \psi \, \mathrm{d}z = 0$ that*

$$\lim_{\epsilon \to 0} \frac{1}{\epsilon} \left\| b[\rho + \epsilon \psi] - b[\rho] - \epsilon \int \delta_\rho b[\rho](z) \psi(z) \mathrm{d}z \right\| = 0 \,.$$

*Proof.* Let $\psi \in C_c^\infty(\mathbb{R}^d)$ with $\int \psi \, \mathrm{d}z = 0$ and fix $\epsilon > 0$. Any minimizer of $G_c(\rho + \epsilon \psi, x)$ for fixed $\rho$ must satisfy

$$\nabla_x G_c(\rho + \epsilon \psi, b(\rho + \epsilon \psi)) = 0 \,.$$

Differentiating in $\epsilon$, we obtain

$$\int \delta_\rho \nabla_x G_c[\rho + \epsilon \psi, b(\rho + \epsilon \psi)] \psi(z) \, \mathrm{d}z + \nabla_x^2 G_c(\rho + \epsilon \psi, b(\rho + \epsilon \psi)) \int \delta_\rho b[\rho + \epsilon \psi](z) \psi(z) \, \mathrm{d}z = 0 \,. \quad (22)$$

Next, we explicitly compute all terms involved in (22). Computing the derivatives yields

$$\nabla_x G_c(\rho, x) = \int \nabla_x f_1(z, x) \mathrm{d}\rho(z) + \int \nabla_x f_2(z, x) \mathrm{d}\bar{\rho}(z) + \beta(x - x_0)$$

$$\delta_\rho \nabla_x G_c[\rho, x](z) = \nabla_x f_1(z, x)$$

$$\nabla_x^2 G_c(\rho, x) = \int \nabla_x^2 f_1(z, x) \mathrm{d}\rho(z) + \int \nabla_x^2 f_2(z, x) \mathrm{d}\bar{\rho}(z) + \beta \, \mathrm{I}_d \,.$$

Note that $\nabla_x^2 G_c$ is invertible by Assumption 1, which states that $f_1$ and $f_2$ have positive-definite Hessians. Inverting this term and substituting these expressions into (22) for $\epsilon = 0$ gives

$$\int \delta_\rho b[\rho](z) \psi(z) \, \mathrm{d}z = - \left[ \beta \, \mathrm{I}_d + \int \nabla_x^2 f_1(z, b(\rho)) \mathrm{d}\rho(z) + \int \nabla_x^2 f_2(z, b(\rho)) \mathrm{d}\bar{\rho}(z) \right]^{-1} \int \nabla_x f_1(z, b(\rho)) \psi(z) \, \mathrm{d}z$$

$$= - \int Q(\rho)^{-1} \nabla_x f_1(z, b(\rho)) \psi(z) \, \mathrm{d}z \,.$$

Finally, the lower bound on $Q(\rho)$ follows thanks to Assumption 1. $\square$

**Remark 9.** *If we include the additional assumption that $f_i \in C^3(\mathbb{R}^d \times \mathbb{R}^d; [0, \infty))$ for $i = 1, 2$, then the Hessian of $b[\rho]$ is well-defined. More precisely, the Hessian is given by*

$$\frac{\mathrm{d}^2}{\mathrm{d}\epsilon^2} b[\rho + \epsilon \psi]|_{\epsilon=0} = Q(\rho)^{-1} \left( \frac{\mathrm{d}}{\mathrm{d}\epsilon} Q(\rho + \epsilon \psi)|_{\epsilon=0} + \int \nabla_x^2 f_1(z, b[\rho]) \psi(z) \mathrm{d}z \right) Q(\rho)^{-1} u[\rho, \psi]$$

*where $u[\rho, \psi] = \int \nabla_x f_1(z, b[\rho]) \psi(z) \mathrm{d}z$ and*

$$\frac{\mathrm{d}}{\mathrm{d}\epsilon} Q_{ij}(\rho + \epsilon \psi)|_{\epsilon=0} = \int \partial_{x_i} \partial_{x_j} f_1(z, b[\rho]) \psi(z) \mathrm{d}z - \int \partial_{x_i} \partial_{x_j} \nabla_x f_1(z, b[\rho]) \psi(z) \rho(z) \mathrm{d}z \, Q(\rho)^{-1} u[\rho, \psi]$$

$$- \int \partial_{x_i} \partial_{x_j} \nabla_x f_2(z, b[\rho]) \psi(z) \bar{\rho}(z) \mathrm{d}z \, Q(\rho)^{-1} u[\rho, \psi].$$

*Therefore, we can Taylor expand $b[\rho]$ up to second order and control the remainder term of order $\epsilon^2$.*

**Lemma 17** (First variation of $G_b$). *The first variation of $G_b$ is given by*

$$\delta_\rho G_b[\rho](z) = h_1(z) + h_2(z) + \beta h_3(z) - \delta_\rho P[\rho](z),$$

*where*

$$h_1(z) := \frac{\delta}{\delta\rho}\left(\int f_1(\tilde{z}, b(\rho))\mathrm{d}\rho(\tilde{z})\right)(z) = \left\langle \int \nabla_x f_1(\tilde{z}, b(\rho))\mathrm{d}\rho(\tilde{z}), \frac{\delta b}{\delta\rho}[\rho](z)\right\rangle + f_1(z, b(\rho)),$$

$$h_2(z) := \frac{\delta}{\delta\rho}\left(\int f_2(\tilde{z}, b(\rho))\mathrm{d}\bar{\rho}(\tilde{z})\right)(z) = \left\langle \int \nabla_x f_2(\tilde{z}, b(\rho))\mathrm{d}\bar{\rho}(\tilde{z}), \frac{\delta b}{\delta\rho}[\rho](z)\right\rangle,$$

$$h_3(z) := \frac{1}{2}\frac{\delta}{\delta\rho}\|b(\rho) - x_0\|^2 = \left\langle b(\rho) - x_0, \frac{\delta b}{\delta\rho}[\rho](z)\right\rangle,$$

*and*

$$\delta_\rho P[\rho](z) = \alpha \log(\rho(z)/\tilde{\rho}(z)) + (W * \rho)(z).$$

*Proof.* We begin with general expressions for Taylor expansions of $b : \mathcal{P}(\mathbb{R}^d) \to \mathbb{R}^d$ and $f_i(z, b(\cdot)) : \mathcal{P}(\mathbb{R}^d) \to \mathbb{R}$ for $i = 1, 2$ around $\rho$. Let $\psi \in \mathcal{T}$ with $\mathcal{T} = \{\psi : \int \psi(z)\mathrm{d}z = 0\}$. Then

$$b(\rho + \epsilon\psi) = b(\rho) + \epsilon \int \frac{\delta b}{\delta\rho}[\rho](z')\psi(z')\mathrm{d}z' + O(\epsilon^2) \tag{23}$$

and

$$f_i(z, b(\rho + \epsilon\psi)) = f_i(z, b(\rho)) + \epsilon\left\langle \nabla_x f_i(z, b(\rho)), \int \frac{\delta b}{\delta\rho}[\rho](z')\psi(z')\mathrm{d}z'\right\rangle + O(\epsilon^2). \tag{24}$$

We compute explicitly each of the first variations:

(i) Using (24), we have

$$\int \psi(z)h_1(z)\mathrm{d}z = \lim_{\epsilon\to 0}\frac{1}{\epsilon}\left[\int f_1(z, b(\rho + \epsilon\psi))(\rho(z) + \epsilon\psi(z))\mathrm{d}z - \int f_1(z, b(\rho))\rho(z)\mathrm{d}z\right]$$

$$= \left\langle \int \nabla_x f_1(z, b(\rho))\mathrm{d}\rho(z), \int \frac{\delta b(\rho)}{\delta\rho}[\rho](z')\psi(z')\mathrm{d}z'\right\rangle + \int f_1(z, b(\rho))\psi(z)\mathrm{d}z$$

$$= \int \left\langle \int \nabla_x f_1(z, b(\rho))\mathrm{d}\rho(z), \frac{\delta b(\rho)}{\delta\rho}[\rho](z')\right\rangle \psi(z')\mathrm{d}z' + \int f_1(z, b(\rho))\psi(z)\mathrm{d}z$$

$$\Rightarrow h_1(z) = \left\langle \int \nabla_x f_1(\tilde{z}, b(\rho))\mathrm{d}\rho(\tilde{z}), \frac{\delta b}{\delta\rho}[\rho](z)\right\rangle + f_1(z, b(\rho)).$$

(ii) Similarly, using again (24),

$$\int \psi(z)h_2(z)\mathrm{d}z = \lim_{\epsilon\to 0}\frac{1}{\epsilon}\left[\int f_2(z, b(\rho + \epsilon\psi))\mathrm{d}\bar{\rho}(z) - \int f_2(z, b(\rho))\bar{\rho}(z)\mathrm{d}z\right]$$

$$= \int \left\langle \int \nabla_x f_2(\tilde{z}, b(\rho))\mathrm{d}\bar{\rho}(\tilde{z}), \frac{\delta b}{\delta\rho}[\rho](z)\right\rangle \psi(z)\mathrm{d}z$$

$$\Rightarrow h_2(z) = \left\langle \int \nabla_x f_2(\tilde{z}, b(\rho))\mathrm{d}\bar{\rho}(\tilde{z}), \frac{\delta b}{\delta\rho}[\rho](z)\right\rangle.$$

(iii) Finally, from (23) it follows that

$$\int \psi(z)h_3(z)\mathrm{d}z = \lim_{\epsilon\to 0}\frac{1}{2\epsilon}\left[\langle b(\rho + \epsilon\psi) - x_0, b(\rho + \epsilon\psi) - x_0\rangle - \langle b(\rho) - x_0, b(\rho) - x_0\rangle\right]$$

$$= \int \left\langle b(\rho) - x_0, \frac{\delta b}{\delta\rho}[\rho](z)\right\rangle \psi(z)\mathrm{d}z$$

$$\Rightarrow h_3(z) = \left\langle b(\rho) - x_0, \frac{\delta b}{\delta\rho}[\rho](z)\right\rangle.$$

Finally, the expression for $\delta_\rho P[\rho]$ follows by direct computation $\qquad\square$

**Lemma 18.** *Denote $G_b(\rho) := G_c(\rho, b(\rho))$ with $b(\rho)$ given by (5). Then $\delta_\rho G_b[\rho] = \delta_\rho G_c[\rho]|_{x=b(\rho)}$.*

*Proof.* We start by computing $\delta_\rho G_c(\cdot, x)[\rho](z)$ for any $z, x \in \mathbb{R}^d$:

$$\delta_\rho G_c(\cdot, x)[\rho](z) = f_1(z, x) - \delta_\rho P[\rho](z). \tag{25}$$

Next, we compute $\delta_\rho G_b$. Using Lemma 17, the first variation of $G_b$ is given by

$$\delta_\rho G_b[\rho](z) = h_1(z) + h_2(z) + \beta h_3(z) - \delta_\rho P[\rho](z)$$
$$= -\left\langle \left[\int \nabla_x f_1(\tilde{z}, b(\rho)) \mathrm{d}\rho(\tilde{z}) + \int \nabla_x f_2(\tilde{z}, b(\rho)) \mathrm{d}\bar{\rho}(\tilde{z}) + \beta(b(\rho) - x_0)\right], \delta_\rho b[\rho](z) \right\rangle$$
$$+ f_1(z, b(\rho)) - \delta_\rho P[\rho](z).$$

Note that

$$\nabla_x G_c(\rho, x) = \int \nabla_x f_1(\tilde{z}, x) \mathrm{d}\rho(\tilde{z}) + \int \nabla_x f_2(\tilde{z}, x) \mathrm{d}\bar{\rho}(\tilde{z}) + \beta(x - x_0), \tag{26}$$

and by the definition of the best response $b(\rho)$, we have $\nabla_x G_x(\rho, x)|_{x=b(\rho)} = 0$. Substituting into the expression for $\delta_\rho G_b$ and using (25), we obtain

$$\delta_\rho G_b[\rho](z) = f_1(z, b(\rho)) - \delta_\rho P[\rho](z) = \delta_\rho G_c(\cdot, x)[\rho](z)\Big|_{x=b(\rho)}.$$

This concludes the proof. $\qquad\square$

**Lemma 19** (Uniform boundedness of the best response). *Let Assumption 1 hold. Then for any $\rho \in \mathcal{P}(\mathbb{R}^d)$, we have*

$$\|b(\rho)\|^2 \leq \|x_0\|^2 + \frac{2(a_1 + a_2)}{\beta}.$$

*Proof.* By definition of the best response $b(\rho)$, we have

$$\int \nabla_x f_1(z, b(\rho)) \mathrm{d}\rho_t + \int \nabla_x f_2(z, b(\rho)) \mathrm{d}\bar{\rho}(z) + \beta(b(\rho) - x_0) = 0.$$

To show that that $b(\rho)$ is uniformly bounded, we take the inner product of the above expression with $b(\rho)$ itself

$$\beta\|b(\rho)\|^2 = \beta x_0 \cdot b(\rho) - \int \nabla_x f_1(z, b(\rho)) \cdot b(\rho) \mathrm{d}\rho(z) - \int \nabla_x f_2(z, b(\rho)) \cdot b(\rho) \mathrm{d}\bar{\rho}(z).$$

Using Assumption 1 to bound the two integrals, together with using Young's inequality to bound the first term on the right-hand side, we obtain

$$\beta\|b(\rho)\|^2 \leq \frac{\beta}{2}\|x_0\|^2 + \frac{\beta}{2}\|b(\rho)\| + a_1 + a_2,$$

which concludes the proof after rearranging terms. $\qquad\square$

**Lemma 20** (Upper semi-continuity). *Let Assumptions 1-3 hold. The functional $G_c : \mathcal{P}(\mathbb{R}^d) \times \mathbb{R}^d \to [-\infty, +\infty]$ is upper semi-continuous when $\mathcal{P}(\mathbb{R}^d) \times \mathbb{R}^d$ is endowed with the product topology of the weak-$*$ topology and the Euclidean topology. Moreover, the functional $G_b : \mathcal{P}(\mathbb{R}^d) \to [-\infty, +\infty]$ is upper semi-continuous with respect to the weak-$*$ topology.*

*Proof.* The functional $G_c : \mathcal{P}(\mathbb{R}^d) \times \mathbb{R}^d \to [-\infty, +\infty]$ is continuous in the second variable thanks to Assumption 1. Similarly, $\int f_1(z, x) \mathrm{d}\rho(z) + \int f_2(z, x) \mathrm{d}\bar{\rho}(z)$ is continuous in $\rho$ thanks to [San15, Proposition 7.1] using the continuity of $f_1$ and $f_2$. Further, $-P$ is upper semi-continuous using [Pos75] and [San15, Proposition 7.2] thanks to Assumptions 2 and 3. This concludes the continuity properties for $G_c$.

The upper semi-continuity of $G_b$ then follows from a direct application of a version of Berge's maximum theorem [AB06, Lemma 16.30]. Let $R := \|x_0\|^2 + \frac{2(a_1+a_2)}{\beta} > 0$. We define $\varphi : (\mathcal{P}(\mathbb{R}^d), W_2) \twoheadrightarrow \mathbb{R}^d$ as the correspondence that maps any $\rho \in \mathcal{P}(\mathbb{R}^d)$ to the closed ball $\overline{B_R(0)} \subset \mathbb{R}^d$. Then the graph of $\varphi$ is $\mathrm{Gr}\,\varphi = \mathcal{P}(\mathbb{R}^d) \times \{\overline{B_R(0)}\}$. With this definition of $\varphi$, the range of $\varphi$ is compact and $\varphi$ is continuous with respect to weak-$*$ convergence, and so it is in particular upper hemicontinuous. Thanks to Lemma 19, the best response function $b(\rho)$ is always contained in $\overline{B_R(0)}$ for any choice of $\rho \in \mathcal{P}(\mathbb{R}^d)$. As a result, maximizing $-G_c(\rho, x)$ in $x$ over $\mathbb{R}^d$ for a fixed $\rho \in \mathcal{P}(\mathbb{R}^d)$ reduces to maximizing it over $\overline{B_R(0)}$. Using the notation introduced above, we can restrict $G_c$ to $G_c : \mathrm{Gr}\,\varphi \to \mathbb{R}$ and write

$$G_b(\rho) := \max_{\hat{x} \in \varphi(\rho)} -G_c(\rho, \hat{x}).$$

Because $G_c(\rho, x)$ is upper semi-continuous when $\mathcal{P}(\mathbb{R}^2) \times \mathbb{R}^d$ is endowed with the product topology of the weak-$*$ topology and the Euclidean topology, [AB06, Lemma 16.30] guarantees that $G_b(\cdot)$ is upper semi-continuous in the weak-$*$ topology. $\qquad\square$

**Proposition 21.** *Let $\alpha, \beta > 0$ and assume Assumptions 1-4 hold with the parameters satisfying $\alpha\tilde{\lambda} > \Lambda_1$. Fix $\rho_0, \rho_1 \in \mathcal{P}(\mathbb{R}^d)$. Along any geodesic $(\rho_s)_{s\in[0,1]} \in \mathcal{P}_2(\mathbb{R}^d)$ connecting $\rho_0$ to $\rho_1$, we have for all $s \in [0,1]$*

$$\frac{\mathrm{d}^2}{\mathrm{d}s^2} G_b(\rho_s) \leq -\lambda_b W_1(\rho_0, \rho_1)^2, \qquad \lambda_b := \alpha\tilde{\lambda} - \Lambda_1, . \tag{27}$$

*As a result, the functional $G_b : \mathcal{P}_2(\mathbb{R}^d) \to [-\infty, +\infty]$ is uniformly displacement concave with constant $\lambda_b > 0$.*

*Proof.* Consider any $\rho_0, \rho_1 \in \mathcal{P}_2(\mathbb{R}^d)$. Then any $W_2$-geodesic $(\rho_s)_{s\in[0,1]}$ connecting $\rho_0$ with $\rho_1$ solves the following system of geodesic equations:

$$\begin{cases} \partial_s \rho_s + \mathrm{div}\,(\rho_s v_s) = 0, \\ \partial_s(\rho_s v_s) + \mathrm{div}\,(\rho_s v_s \otimes v_s) = 0, \end{cases} \tag{28}$$

where $\rho_s : \mathbb{R}^d \to \mathbb{R}$ and $v_s : \mathbb{R}^d \mapsto \mathbb{R}^d$. The first derivative of $G_b$ along geodesics can be computed explicitly as

$$\frac{\mathrm{d}}{\mathrm{d}s} G_b(\rho_s) = \int \nabla_z f_1(z, b(\rho_s)) \cdot v_s(z)\rho_s(z)\mathrm{d}z - \frac{\mathrm{d}}{\mathrm{d}s}P(\rho_s)$$
$$+ \left\langle \left[\int \nabla_x f_1(z, x)\mathrm{d}\rho_s(z) + \int \nabla_x f_2(z, x)\mathrm{d}\bar{\rho}(z) + \beta(x - x_0)\right]\Big|_{x=b(\rho_s)}, \frac{\mathrm{d}}{\mathrm{d}s}b(\rho_s)\right\rangle.$$

The left-hand side of the inner product is zero by definition of the best response $b(\rho_s)$ to $\rho_s$, see (26). Therefore

$$\frac{\mathrm{d}}{\mathrm{d}s} G_b(\rho_s) = \int \nabla_z f_1(z, b(\rho_s)) \cdot v_s(z)\rho_s(z)\mathrm{d}z - \frac{\mathrm{d}}{\mathrm{d}s}P(\rho_s).$$

Differentiating a second time, using (28) and integration by parts, we obtain

$$\frac{\mathrm{d}^2}{\mathrm{d}s^2} G_b(\rho_s) = L_1(\rho_s) + L_2(\rho_s) - \frac{\mathrm{d}^2}{\mathrm{d}s^2}P(\rho_s),$$

where

$$L_1(\rho_s) := \int \nabla_z^2 f_1(z, b(\rho_s)) \cdot (v_s \otimes v_s)\,\rho_s \mathrm{d}z = \int \left\langle v_s, \nabla_z^2 f_1(z, b(\rho_s)) \cdot v_s \right\rangle \rho_s \mathrm{d}z,$$

$$L_2(\rho_s) := \int \frac{\mathrm{d}}{\mathrm{d}s}b(\rho_s) \cdot \nabla_x \nabla_z f_1(z, b(\rho_s)) \cdot v_s(z)\,\rho_s(z)\mathrm{d}z.$$

From (16), we have that

$$\frac{\mathrm{d}^2}{\mathrm{d}s^2}\tilde{P}(\rho_s) \geq \alpha\tilde{\lambda}\,W_2(\rho_0, \rho_1)^2,$$

and thanks to Assumption 4 it follows that

$$L_1(s) \leq \Lambda_1 W_2(\rho_0, \rho_1)^2.$$

This leaves $L_2$ to bound; we first consider the term $\frac{\mathrm{d}}{\mathrm{d}s}b(\rho_s)$:

$$\frac{\mathrm{d}}{\mathrm{d}s}b(\rho_s) = \int \delta_\rho b[\rho_s](\tilde{z})\partial_s \rho_s(\mathrm{d}\tilde{z}) = -\int \delta_\rho b[\rho_s](\tilde{z})\mathrm{div}\,(\rho_s v_s)\,\mathrm{d}\tilde{z}$$
$$= \int \nabla_z \delta_\rho b[\rho_s](\tilde{z}) \cdot v_s(\tilde{z})\mathrm{d}\rho_s(\tilde{z}).$$

Defining $u(\rho_s) \in \mathbb{R}^d$ by

$$u(\rho_s) := \int \nabla_x \nabla_z f_1(z, b(\rho_s)) \cdot v_s(z)\mathrm{d}\rho_s(z),$$

using the results from Lemma 16 for $\nabla_z \delta_\rho b[\rho_s]$, Assumption 1 and the fact that $Q(\rho)$ is constant in $z$ and $x$, we have

$$L_2(\rho_s) = -\iint \left[Q(\rho_s)^{-1}\nabla_x \nabla_z f_1(\tilde{z}, b(\rho_s)) \cdot v_s(\tilde{z})\right] \cdot \nabla_x \nabla_z f_1(z, b(\rho_s)) \cdot v_s(z)\,\mathrm{d}\rho_s(z)\mathrm{d}\rho_s(\tilde{z})$$
$$= -\left\langle u(\rho_s), Q(\rho_s)^{-1}u(\rho_s)\right\rangle \leq 0$$

Combining all terms together, we obtain

$$\frac{\mathrm{d}^2}{\mathrm{d}s^2} G_b(\rho_s) \leq -\left(\alpha\tilde{\lambda} - \Lambda_1\right) W_2(\rho_0, \rho_1)^2.$$

$\square$

**Remark 10.** *Under some additional assumptions on the functions $f_1$ and $f_2$, we can obtain an improved convergence rate. In particular, assume that for all $z, x \in \mathbb{R}^d$,*

- *there exists a constant $\Lambda_2 \geq \lambda_2 \geq 0$ such that $\nabla_x^2 f_2(z, x) \preceq \Lambda_2 \, \mathrm{I}_d$;*

- *there exists a constant $\sigma \geq 0$ such that $\|\nabla_x \nabla_z f_1(z, x)\| \geq \sigma$.*

*Then we have $-Q(\rho_s)^{-1} \preceq -1/(\beta + \Lambda_1 + \Lambda_2) \, \mathrm{I}_d$. Using Lemma 16, we then obtain a stronger bound on $L_2$ as follows:*

$$
L_2(\rho_s) \leq -\frac{1}{\beta + \Lambda_1 + \Lambda_2} \|u(\rho_s)\|^2 \leq -\frac{1}{\beta + \Lambda_1 + \Lambda_2} \int \|\nabla_x \nabla_z f_1(z, b(\rho_s))\|^2 \, \mathrm{d}\rho_s(z) \int \|v_s(z)\|^2 \, \mathrm{d}\rho_s(z)
$$

$$
\leq -\frac{\sigma^2}{\beta + \Lambda_1 + \Lambda_2} W_2(\rho_0, \rho_1)^2 .
$$

*This means we can improve the convergence rate in (27) to $\lambda_b := \alpha\tilde{\lambda} + \frac{\sigma^2}{\beta + \Lambda_1 + \Lambda_2} - \Lambda_1$.*

**Proposition 22** (Ground state). *Let Assumptions 1-4 hold for $\alpha\tilde{\lambda} > \Lambda_1 \geq 0$ and $\beta > 0$. Then there exists a unique maximizer $\rho_*$ for the functional $G_b$ over $\mathcal{P}(\mathbb{R}^d)$, and it satisfies $\rho_* \in \mathcal{P}_2(\mathbb{R}^d) \cap L^1(\mathbb{R}^d)$ and $\rho_*$ is absolutely continuous with respect to $\tilde{\rho}$.*

*Proof.* Uniqueness of the maximizer (if it exists) is guaranteed by the uniform concavity provided by Lemma 21. To show existence of a maximizer, we use the direct method in the calculus of variations, requiring the following key properties for $G_b$: (1) boundedness from above, (2) upper semi-continuity, and (3) tightness of any minimizing sequence. To show (1), note that $\nabla_z^2 (f_1(z, x) + \alpha \log \tilde{\rho}(z)) \preceq -(\alpha\tilde{\lambda} - \Lambda_1) \, \mathrm{I}_d$ for all $z, x \in \mathbb{R}^d \times \mathbb{R}^d$ by Assumptions 2 and 4, and so

$$
f_1(z, x) + \alpha \log \tilde{\rho}(z) \leq c_0(x) - \frac{(\alpha\tilde{\lambda} - \Lambda_1)}{4}|z|^2 \qquad \forall (z, x) \in \mathbb{R}^d \times \mathbb{R}^d \tag{29}
$$

with $c_0(x) := f_1(0, x) + \alpha \log \tilde{\rho}(0) + \frac{1}{\alpha\tilde{\lambda} - \Lambda_1} \|\nabla_z \left[ f_1(0, x) + \alpha \log \tilde{\rho}(0) \right] \|^2$. Therefore,

$$
G_b(\rho) = \int \left[ f_1(z, b(\rho)) + \alpha \log \tilde{\rho}(z) \right] \mathrm{d}\rho(z) + \int f_2(z, b(\rho)) \mathrm{d}\bar{\rho}(z) + \frac{\beta}{2} \|b(\rho) - x_0\|^2
$$

$$
- \alpha \int \rho \log \rho - \int \rho W * \rho
$$

$$
\leq c_0(b(\rho)) + \int f_2(z, b(\rho)) \mathrm{d}\bar{\rho}(z) + \frac{\beta}{2} \|b(\rho) - x_0\|^2 .
$$

To estimate each of the remaining terms on the right-hand side, denote $R := \|x_0\|^2 + \frac{2(a_1 + a_2)}{\beta}$ and recall that $\|b(\rho)\| \leq R$ for any $\rho \in \mathcal{P}(\mathbb{R}^d)$ thanks to Lemma 19. By continuity of $f_1$ and $\log \tilde{\rho}$, there exists a constant $c_1 \in \mathbb{R}$ such that

$$
\sup_{x \in B_R(0)} c_0(x) = \sup_{x \in B_R(0)} \left[ f_1(0, x) + \alpha \log \tilde{\rho}(0) + \frac{1}{\alpha\tilde{\lambda} - \Lambda_1} \|\nabla_z \left( f_1(0, x) + \alpha \log \tilde{\rho}(0) \right) \|^2 \right] \leq c_1 . \tag{30}
$$

The second term is controlled by $c_2$ thanks to Assumption 4. And the third term can be bounded directly to obtain

$$
G_b(\rho) \leq c_1 + c_2 + \beta (R^2 + \|x_0\|^2) .
$$

This concludes the proof of (1). Statement (2) was shown in Lemma 20. Then we obtain a minimizing sequence $(\rho_n) \in \mathcal{P}(\mathbb{R}^d)$ which is in the closed unit ball of $C_0(\mathbb{R}^d)^*$ and so the Banach-Anaoglu theorem [Rud91, Theorem 3.15] there exists a limit $\rho_*$ in the Radon measures and a subsequence (not relabeled) such that $\rho_n \overset{*}{\rightharpoonup} \rho_*$. In fact, $\rho_*$ is absolutely continuous with respect to $\tilde{\rho}$ as otherwise $G_b(\rho_*) = -\infty$, which contradicts that $G_b(\cdot) > -\infty$ somewhere. We conclude that $\rho_* \in L^1(\mathbb{R}^d)$ since $\tilde{\rho} \in L^1(\mathbb{R}^d)$ by Assumption 2. To ensure $\rho_* \in \mathcal{P}(\mathbb{R}^d)$, we require (3) tightness of the minimizing sequence $(\rho_n)$. By Markov's inequality [Gho02] it is sufficient to establish a uniform bound on the second moments:

$$
\int \|z\|^2 \mathrm{d}\rho_n(z) < C \qquad \forall n \in \mathbb{N} . \tag{31}
$$

To see this we proceed in a similar way as in the proof of Proposition 10. Defining

$$
K(\rho) := -\int \left[ f_1(z, b(\rho)) + \alpha \log \tilde{\rho}(z) \right] \mathrm{d}\rho(z) + \alpha \int \rho \log \rho \, \mathrm{d}z + \frac{1}{2} \int \rho W * \rho \, \mathrm{d}z ,
$$

we have $K(\rho) = -G_b(\rho) + \int f_2(z, b(\rho)) \mathrm{d}\bar{\rho}(z) + \frac{\beta}{2} \|b(\rho) - x_0\|^2$. Then using again the bound on $b(\rho)$ from Lemma 19,

$$K(\rho) \leq -G_b(\rho) + \sup_{x \in B_R(0)} \int f_2(z, x) \mathrm{d}\bar{\rho}(z) + \beta \left(R^2 + \|x_0\|^2\right)$$
$$\leq -G_b(\rho) + c_2 + \beta \left(R^2 + \|x_0\|^2\right),$$

where the last inequality is thanks to Assumption 4. Hence, using the estimates (29) and (30) from above, and noting that the sequence $(\rho_n)$ is minimizing $(-G_b)$, we have

$$\frac{(\alpha\tilde{\lambda} - \Lambda_1)}{4} \int \|z\|^2 \, \mathrm{d}\rho_n(z) \leq c_0(b(\rho_n)) - \int \left[f_1(z, b(\rho_n)) + \alpha \log \tilde{\rho}(z)\right] \mathrm{d}\rho_n(z)$$
$$\leq c_1 + K(\rho_n) \leq c_1 - G_b(\rho_n) + c_2 + \beta \left(R^2 + \|x_0\|^2\right)$$
$$\leq c_1 - G_b(\rho_1) + c_2 + \beta \left(R^2 + \|x_0\|^2\right) < \infty.$$

which uniformly bounds the second moments of $(\rho_n)$. This concludes the proof for the estimate (31) and also ensures that $\rho_* \in \mathcal{P}_2(\mathbb{R}^d)$. $\qquad\square$

**Corollary 23.** *Any maximizer $\rho_*$ of $G_b$ is a steady state for equation (5) according to Definition 9, and satisfies* $\mathrm{supp}(\rho_*) = \mathrm{supp}(\tilde{\rho})$.

*Proof.* To show that $\rho_*$ is a steady state we can follow exactly the same argument as in the proof of Corollary 12, just replacing $-\frac{1}{\alpha}\int f_1(z, x) \, \mu_*(x)$ with $+\frac{1}{\alpha}\int f_1(z, b(\rho_*))$. It remains to show that $\mathrm{supp}(\rho_*) = \mathrm{supp}(\tilde{\rho})$. As $\rho^*$ is a maximizer, it is in particular a critical point, and therefore satisfies that $\delta_\rho G_b[\rho_*](z)$ is constant on all connected components of $\mathrm{supp}(\rho_*)$. Thanks to Lemma 18, this means there exists a constant $c[\rho_*]$ (which may be different on different components of $\mathrm{supp}(\rho_*)$) such that

$$f_1(z, b(\rho_*)) - \alpha \log\left(\frac{\rho_*(z)}{\tilde{\rho}(z)}\right) - W * \rho_*(z) = c[\rho_*] \qquad \text{on } \mathrm{supp}(\rho_*).$$

Rearranging, we obtain (for a possible different constant $c[\rho_*] \neq 0$)

$$\rho_*(z) = c[\rho_*]\tilde{\rho}(z) \exp\left[\frac{1}{\alpha}\left(f_1(z, b(\rho_*)) - W * \rho_*(z)\right)\right] \qquad \text{on } \mathrm{supp}(\rho_*). \tag{32}$$

Firstly, note that $\mathrm{supp}(\rho_*) \subset \mathrm{supp}(\tilde{\rho})$ since $\rho_*$ is absolutely continuous with respect to $\tilde{\rho}$. Secondly, note that $\exp \frac{1}{\alpha} f_1(z, b(\rho_*)) \geq 1$ for all $z \in \mathbb{R}^d$ since $f_1 \geq 0$. Finally, we claim that $\exp\left(-\frac{1}{\alpha} W * \rho_*(z)\right) > 0$ for all $z \in \mathbb{R}^d$. In other words, we claim that $W * \rho_*(z) < \infty$ for all $z \in \mathbb{R}^d$. This follows by exactly the same argument as in Corollary 12, see equation (19). We conclude that $\mathrm{supp}(\rho_*) = \mathrm{supp}(\tilde{\rho})$. $\qquad\square$

**Remark 11.** *If we have in addition that $\tilde{\rho} \in L^\infty(\mathbb{R}^d)$ and $f_1(\cdot, x) \in L^\infty(\mathbb{R}^d)$ for all $x \in \mathbb{R}^d$, then the maximizer $\rho_*$ of $G_b$ is in $L^\infty(\mathbb{R}^d)$ as well. This follows directly by bounding the right-hand side of (32).*

With the above preliminary results, we can now show the HWI inequality, which implies again a Talagrand-type inequality and a generalized logarithmic Sobolev inequality.

**Proposition 24** (HWI inequalities). *Define the dissipation functional*

$$D_b(\gamma) := \iint |\nabla_z \delta_\rho G_b[\rho](z)|^2 \mathrm{d}\rho(z).$$

*Assume $\alpha, \beta > 0$ such that $\alpha\tilde{\lambda} > \Lambda_1 + \sigma^2$, and let $\lambda_b$ as defined in (27). Denote by $\rho_*$ the unique maximizer of $G_b$.*

*(HWI) Let $\rho_0, \rho_1 \in \mathcal{P}_2(\mathbb{R}^d)$ such that $G_b(\rho_0), G_b(\rho_1), D_b(\rho_0) < \infty$. Then*

$$G_b(\rho_0) - G_b(\rho_1) \leq \overline{W}(\rho_0, \rho_1)\sqrt{D_b(\rho_0)} - \frac{\lambda_b}{2} W_2(\rho_0, \rho_1)^2 \tag{33}$$

*(logSobolev) Any $\rho \in \mathcal{P}_2(\mathbb{R}^d)$ such that $G(\rho), D_b(\rho) < \infty$ satisfies*

$$D_b(\rho) \geq 2\lambda_b \, G_a(\rho|\rho_*). \tag{34}$$

*(Talagrand) For any $\rho \in \mathcal{P}_2(\mathbb{R}^d)$ such that $G_b(\rho) < \infty$, we have*

$$W_2(\rho, \rho_*)^2 \leq \frac{2}{\lambda_b} G_b(\rho \,|\, \rho_*). \tag{35}$$

*Proof.* The proof for this result follows analogously to the arguments presented in the proofs of Proposition 13, Corollary 14 and Corollary 15, using the preliminary results established in Proposition 21 and Proposition 22. $\square$

*Proof of Theorem 3.* Following the same approach as in the proof of Theorem 2, the results in Theorem 3 immediately follow by combining Proposition 22, Corollary 23 and Proposition 24 applied to solutions of the PDE (5). $\square$

# D   Proof of Theorem 4

The proof for this theorem uses similar strategies as that of Theorem 3, but considers the evolution of an ODE rather than a PDE. Recall that for any $x \in \mathbb{R}^d$ the best response $r(x)(\cdot) \in \mathcal{P}(\mathbb{R}^d)$ in (6) is defined as

$$r(x) := \operatorname*{argmax}_{\hat{\rho} \in \mathcal{P}} G_c(\hat{\rho}, x),$$

where the energy $G_c(\rho, x) : \mathcal{P}(\mathbb{R}^d) \times \mathbb{R}^d \mapsto [-\infty, \infty]$ is given by

$$G_c(\rho, x) = \int f_1(z, x) \mathrm{d}\rho(z) + \int f_2(z, x) \mathrm{d}\bar{\rho}(z) + \frac{\beta}{2} \|x - x_0\|^2 - \alpha KL(\rho|\tilde{\rho}) - \frac{1}{2} \int \rho W * \rho.$$

**Lemma 25.** *Let Assumptions 2- 4 hold and assume $\alpha \tilde{\lambda} > \Lambda_1$. Then for each $x \in \mathbb{R}^d$ there exists a unique maximizer $\rho_* := r(x)$ solving $\operatorname{argmax}_{\hat{\rho} \in \mathcal{P}_2} G_c(\hat{\rho}, x)$. Further, $r(x) \in L^1(\mathbb{R}^d)$, $\operatorname{supp}(r(x)) = \operatorname{supp}(\tilde{\rho})$, and there exists a function $c : \mathbb{R}^d \mapsto \mathbb{R}$ such that the best response $\rho_*(z) = r(x)(z)$ solves the Euler-Lagrange equation*

$$\delta_\rho G_c[\rho_*, x](z) := \alpha \log \rho_*(z) - (f_1(z, x) + \alpha \log \tilde{\rho}(z)) + (W * \rho_*)(z) = c(x) \ \ for \ all \ (z, x) \in \operatorname{supp}(\tilde{\rho}) \times \mathbb{R}^d. \tag{36}$$

*Proof.* Equivalently, consider the minimization problem for $F(\rho) = -\int f_1(z, x) \, \mathrm{d}\rho(z) + \alpha KL(\rho \,|\, \tilde{\rho}) + \frac{1}{2} \int \rho W * \rho$ with some fixed $x$. Note that we can rewrite $F(\rho)$ as

$$F(\rho) = \alpha \int \rho \log \rho \, \mathrm{d}z + \int V(z, x) \mathrm{d}\rho(z) + \frac{1}{2} \int \rho W * \rho$$

where $V(z, x) := -(f_1(z, x) + \alpha \log \tilde{\rho}(z))$ is strictly convex in $z$ for fixed $x$ by Assumptions 2 and 4. Together with Assumption 3, we can directly apply the uniqueness and existence result from [CMV03, Theorem 2.1 (i)].

The result on the support of $r(x)$ and the expression for the Euler-Lagrange equation follows by exactly the same arguments as in Corollary 12 and Corollary 23. $\square$

**Lemma 26.** *The density of the best response $r(x)$ is continuous on $\mathbb{R}^d$ for any fixed $x \in \mathbb{R}^d$.*

*Proof.* Instead of solving the Euler-Lagrange equation (36), we can also obtain the best response $r(x)$ as the long-time asymptotics for the following gradient flow:

$$\partial_t \rho = \operatorname{div}(\rho \nabla \delta_\rho F[\rho]). \tag{37}$$

Following Definitions 8 and 9, we can characterize the steady states $\rho_\infty$ of the PDE (37) by requiring that $\rho_\infty \in L^1_+(\mathbb{R}^d) \cap L^\infty_{loc}(\mathbb{R}^d)$ with $\|\rho_\infty\|_1 = 1$ such that $\rho_\infty \in W^{1,2}_{loc}(\mathbb{R}^d)$, $\nabla W * \rho_\infty \in L^1_{loc}(\mathbb{R}^d)$, $\rho_\infty$ is absolutely continuous with respect to $\tilde{\rho}$, and $\rho_\infty$ satisfies

$$\nabla_z \left( -f_1(z, x) + \alpha \log\left( \frac{\rho_\infty(z)}{\tilde{\rho}(z)} \right) + W * \rho_\infty(z) \right) = 0 \qquad \forall z \in \mathbb{R}^d, \tag{38}$$

in the sense of distributions. Noting that because the energy functional $F(\rho)$ differs from $G_a(\rho, \mu)$ only in the sign of $f_1(z, x)$ if viewing $G_a(\rho, \mu)$ as a function of $\rho$ only. Note that $F(\rho)$ is still uniformly displacement convex in $\rho$ due to Assumption 4. Then the argument to obtain that $\rho_\infty \in C(\mathbb{R}^d)$ follows exactly as that of Lemma 8. $\square$

**Lemma 27.** *Let $i \in \{1, ..., d\}$. If the energy $H_i : \mathcal{P}(\mathbb{R}^d) \to \mathbb{R}^d$ given by*

$$H_i(\rho, x) := \frac{\alpha}{2} \int \frac{\rho(z)^2}{r(x)(z)} \, \mathrm{d}z + \frac{1}{2} \int \rho W * \rho - \int \partial_{x_i} f_1(z, x) \mathrm{d}\rho(z), \tag{39}$$

*admits a critical point at $x \in \mathbb{R}^d$, then the best response $r(x) \in \mathcal{P}(\mathbb{R}^d)$ is differentiable in the ith coordinate direction at $x \in \mathbb{R}^d$. Further, the critical point of $H_i$ is in the subdifferential $\partial_{x_i} r(x)$.*

*Proof.* First, note that $DF[r(x)](x)(u) = 0$ for all directions $u \in C_c^\infty(\mathbb{R}^d)$ and for all $x \in \mathbb{R}^d$ thanks to optimality of $r(x)$. Here, $DF$ denotes the Fréchet derivative of $F$, associating to every $\rho \in \mathcal{P}(\mathbb{R}^d)$ the bounded linear operator $DF[\rho] : C_c^\infty \to \mathbb{R}$

$$DF[\rho](u) := \int \delta_\rho F[\rho](z) u(z) \mathrm{d}z \,,$$

and we note that $F(\rho)$ depends on $x$ through the potential $V$. Fixing an index $i \in \{1, ..., d\}$, and differentiating the optimality condition with respect to $x_i$ we obtain

$$\partial_{x_i} DF[r(x)](x)(u) + D^2 F[r(x)](x)(u, \partial_{x_i} r(x)) = 0 \qquad \forall u \in C_c^\infty(\mathbb{R}^d) \,. \tag{40}$$

Both terms can be made more explicit using the expressions for the Fréchet derivative of $F$:

$$\partial_{x_i} DF[r(x)](x)(u) = -\int \partial_{x_i} f_1(z, x) u(z) \mathrm{d}z \,,$$

and for the second term note that the second Fréchet derivative of $F$ at $\rho \in \mathcal{P}(\mathbb{R}^d)$ along directions $u, v \in C_c^\infty(\mathbb{R}^d)$ such that $(\mathrm{supp}(u) \cup \mathrm{supp}(v)) \subset \mathrm{supp}(\rho)$ is given by

$$D^2 F[\rho](x)(u, v) = \alpha \int \frac{u(z) v(z)}{\rho(z)} \, \mathrm{d}z + \iint W(z - \tilde{z}) u(z) v(\tilde{z}) \, \mathrm{d}z \mathrm{d}\tilde{z} \,.$$

In other words, assuming $\mathrm{supp}(r(x)) = \mathrm{supp}(\tilde{\rho}) = \mathbb{R}^d$, relation (40) can be written as

$$\alpha \int \frac{\partial_{x_i} r(x)}{r(x)(z)} u(z) \, \mathrm{d}z + \int (W * \partial_{x_i} r(x))(z) u(z) v \, \mathrm{d}z - \int \partial_{x_i} f_1(z, x) u(z) \mathrm{d}z = 0 \,,$$

For ease of notation, given $r(x) \in \mathcal{P}(\mathbb{R}^d)$, we define the function $g : \mathcal{P}(\mathbb{R}^d) \to L_{loc}^1(\mathbb{R}^d)$ by

$$g[\rho](z) := \alpha \frac{\rho(z)}{r(x)(z)} + W * \rho - \partial_{x_i} f_1(z, x) \,.$$

The question whether the partial derivative $\partial_{x_i} r(x)$ exists then reduces to the question whether there exists some $\rho_* \in \mathcal{P}(\mathbb{R}^d)$ such that $\rho = \rho_*$ solves the equation

$$g[\rho](z) = c \qquad \text{for almost every } z \in \mathbb{R}^d \,.$$

and for some constant $c > 0$. This is precisely the Euler-Lagrange condition for the functional $H_i$ defined in (39), which has a solution thanks to the assumption of Lemma 27.

$$\square$$

We observe that the first term in $H_i$ is precisely (up to a constant) the $\chi^2$-divergence with respect to $r(x)$,

$$\int \left( \frac{\rho}{r(x)} - 1 \right)^2 r(x) \, \mathrm{d}z = \int \frac{\rho^2}{r(x)} \, \mathrm{d}z - 1 \,.$$

Depending on the shape of the best response $r(x)$, the $\chi^2$-divergence may not be displacement convex. Similarly, the last term $-\int \partial_{x_i} f_1(z, x) \mathrm{d}\rho(z)$ in the energy $H_i$ is in fact displacement concave due to the convexity properties of $f_1$ in $z$. The interaction term is displacement convex thanks to Assumption 3. As a result, the overall convexity properties of $H_i$ are not known in general. Proving the existence of a critical for $H_i$ under our assumptions on $f_1, f_2, \tilde{\rho}$ and $W$ would be an interesting result in its own right, providing a new functional inequality that expands on the literature of related functional inequalities such as the related Hardy-Littlewood-Sobolev inequality [Lie83].

It remains to show that $H_i$ indeed admits a critical point. Next, we provide examples of additional assumptions that would guarantee for Lemma 27 to apply.

**Lemma 28.** *If either $C := \sup_{z \in \mathbb{R}^d} |W(z)| < \infty$, or*

$$C := \sup_{z \in \mathbb{R}^d} |\alpha \log(r(x)(z)/\tilde{\rho}(z)) + f_1(z, x) + c| < \infty \,,$$

*then for each $x \in \mathbb{R}^d$ and for large enough $\alpha > 0$, the best response $r(x)$ is differentiable with the gradient coordinate $\partial_{x_i} r(x)$ given by the unique coordinate-wise solutions of the Euler-Lagrange condition for $H_i$.*

*Proof.* We will show this result using the Banach Fixed Point Theorem for the mapping $T_i : L^1(\mathbb{R}^d) \to L^1(\mathbb{R}^d)$ for each fixed $i \in \{1, ..., d\}$ given by

$$T_i(\rho) = -\frac{r(x)(z)}{\alpha} [(W * \rho)(z) - \partial_{x_i} f_1(z, x) + c] \,,$$

noting that $\rho_* = T_i(\rho_*)$ is the Euler-Lagrange condition for a critical point of $H_i$. It remains to show that $T_i$ is a contractive mapping. For the first assumption, note that

$$\left\|T_i(\rho) - T_i(\rho')\right\|_1 = \frac{1}{\alpha} \int r(x)|W * (\rho - \rho')|\mathrm{d}z$$

$$\leq \frac{1}{\alpha} \iint r(x)(z)W(z - \hat{z})|\rho(\hat{z}) - \rho'(\hat{z})|\mathrm{d}\hat{z}\mathrm{d}z$$

$$\leq \frac{\|W\|_\infty}{\alpha} \left(\int r(x)(z)\mathrm{d}z\right)\left(\int |\rho(\hat{z}) - \rho'(\hat{z})|\mathrm{d}\hat{z}\right) \leq \frac{C}{\alpha} \left\|\rho - \rho'\right\|_1 .$$

Similarly, for the second assumption we estimate

$$\left\|T_i(\rho) - T_i(\rho')\right\|_1 = \frac{1}{\alpha} \int r(x)|W * (\rho - \rho')|\mathrm{d}z$$

$$\leq \frac{1}{\alpha} \iint r(x)(z)W(z - \hat{z})|\rho(\hat{z}) - \rho'(\hat{z})|\mathrm{d}\hat{z}\mathrm{d}z$$

$$= \frac{1}{\alpha} \int (W * r(x))(z)|\rho(z) - \rho'(z)|\mathrm{d}z$$

$$\leq \frac{1}{\alpha} \|W * r(x)\|_\infty \left\|\rho - \rho'\right\|_1$$

which requires a bound on $\|W * r(x)\|_\infty$. Using

$$\|W * r(x)\|_\infty = \sup_{z \in \mathbb{R}^d} |\alpha \log(r(x)/\tilde{\rho}) + f_1(z, x) + c| = C < \infty ,$$

we conclude that $T_i$ is a contraction map for large enough $\alpha$. In both cases, we can then apply the Banach Fixed-Point Theorem to conclude that $\nabla_x r(x)$ exists and is unique. □

**Lemma 29.** *Let $r(x)$ as defined in* (6). *If $r(x)$ is differentiable in $x$, then we have $\nabla_x G_d(x) = (\nabla_x G_c(\rho, x))|_{\rho=r(x)}$.*

*Proof.* We start by computing $\nabla_x G_d(x)$. We have

$$\nabla_x G_d(x) = \nabla_x (G_c(r(x), x)) = \int \delta_\rho[G_c(\rho, x)]|_{\rho=r(x)}(z)\nabla_x r(x)(z)\mathrm{d}z + (\nabla_x G_c(\rho, x))|_{\rho=r(x)}$$

$$= c(x)\nabla_x \int r(x)(z)\mathrm{d}z + (\nabla_x G_c(\rho, x))|_{\rho=r(x)} = (\nabla_x G_c(\rho, x))|_{\rho=r(x)} ,$$

where we used that $r(x)$ solves the Euler-Lagrange equation (36) and that $r(x) \in \mathcal{P}(\mathbb{R}^d)$ for any $x \in \mathbb{R}^d$ so that $\int r(x)(z)\mathrm{d}z$ is independent of $x$. □

**Lemma 30.** *Let Assumption 1 hold. Then $G_d : \mathbb{R}^d \to \mathbb{R} \cup \{+\infty\}$ is strongly convex with constant $\lambda_d := \lambda_1 + \lambda_2 + \beta > 0$.*

*Proof.* The energy $G_c(\rho, x)$ is strongly convex in $x$ due to our assumptions on $f_1$, $f_2$, and the regularizing term $\|x - x_0\|_2^2$. This means that for any $\rho \in \mathcal{P}$,

$$G_c(\rho, x) \geq G_c(\rho, x') + \nabla_x G_c(\rho, x')^\top (x - x') + \frac{\lambda_d}{2} \left\|x - x'\right\|_2^2 .$$

Selecting $\rho = r(x')$, we have

$$G_c(r(x'), x) \geq G_c(r(x'), x') + \nabla_x G_c(r(x'), x')^\top (x - x') + \frac{\lambda_d}{2} \left\|x - x'\right\|_2^2 .$$

Since $G_c(r(x'), x) \leq G_c(r(x), x)$ by definition of $r(x)$, we obtain the required convexity condition:

$$G_d(x) = G_c(r(x), x) \geq G_c(r(x'), x') + \nabla_x G_c(r(x'), x')^\top (x - x') + \frac{\lambda_d}{2} \left\|x - x'\right\|_2^2 .$$

□

*Proof of Theorem 4.* For any reference measure $\rho_0 \in \mathcal{P}$, we have

$$G_d(x) \geq G_c(\rho_0, x) \geq -\alpha KL(\rho_0 \mid \tilde{\rho}) - \frac{1}{2} \int \rho_0 W * \rho_0 + \frac{\beta}{2}\|x - x_0\|^2$$

and therefore, $G_d$ is coercive. Together with the strong convexity provided by Lemma 30, we obtain the existence of a unique minimizer $x_\infty \in \mathbb{R}^d$. Convergence in norm now immediately follows also using Lemma 30: for solutions $x(t)$ to (6), we have

$$\frac{1}{2}\frac{d}{dt}\|x(t) - x_\infty\|^2 = -\left(G_d(x(t)) - G_d(x_\infty)\right)\cdot(x(t) - x_\infty) \leq -\lambda_d\|x(t) - x_\infty\|^2.$$

A similar result holds for convergence in entropy using the Polyák-Łojasiewicz convexity inequality

$$\frac{1}{2}\|\nabla G_d(x)\|_2^2 \geq \lambda_d(G_d(x) - G_d(x_\infty)),$$

which is itself a direct consequence of strong convexity provided in Lemma 30. Then

$$\frac{d}{dt}\left(G_d(x(t)) - G_d(x_\infty)\right) = \nabla_x G_d(x(t))\cdot\dot{x}(t) = -\|\nabla_x G_d(x(t))\|^2 \leq -2\lambda_d\left(G_d(x(t)) - G_d(x_\infty)\right),$$

and so the result in Theorem 4 follows. $\qquad\square$

# E    Additional Simulation Results

We simulate a number of additional scenarios to illustrate extensions beyond the setting with provable guarantees and in the settings for which we have results but no numerical implementations in the main paper. First, we simulate the aligned objectives setting in one dimension, corresponding to (4). Then we consider two settings which are not covered in our theory: (1) the previously-fixed distribution $\bar{\rho}$ is also time varying, and (2) the algorithm does not have access to the full distributions of $\rho$ and $\bar{\rho}$ and instead samples from them to update. Lastly, we illustrate a classifier with the population attributes in two dimensions, which requires a different finite-volume implementation [CCH15, Section 2.2] than the one dimension version of the PDE due to flux in two dimensions.

## E.1    Aligned Objectives

Here we show numerical simulation results for the aligned objectives case, where the population and distribution have the same cost function. In this setting, the dynamics are of the form

$$\partial_t \rho = \text{div}\left(\rho\nabla_z\delta_\rho G_a[\rho,\mu]\right)$$
$$= \text{div}\left(\rho\nabla_z\left(\int f_1(z,x)d\mu(x) + \alpha\log(\rho/\tilde{\rho}) + W * \rho\right)\right)$$
$$\frac{d}{dt}x = -\nabla_x\left(\int f_1(z,x)d\rho(z) + \int f_2(z,x)d\bar{\rho}(z) + \frac{\beta}{2}\|x - x_0\|^2\right)$$

where $f_1$ and $f_2$ are as defined in section 4.1, and $W = \frac{1}{20}(1 + z)^{-1}$, a consensus kernel. Note that $W$ does not satisfy Assumption 3, but we still observe convergence in the simulation. This is expected; in other works such as [CMV03], the assumptions on $W$ are relaxed and convergence results proven given sufficient convexity of other terms. The regularizer $\tilde{\rho}$ is set to $\rho_0$, which models a penalty for the effort required of individuals to alter their attributes. The coefficient weights are $\alpha = 0.1$ and $\beta = 1$, with discretization parameters $dz = 0.1$, $dt = 0.01$.

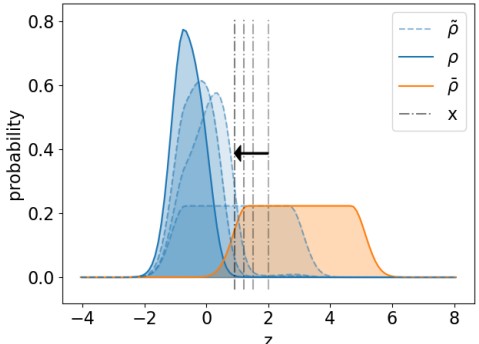

Figure 4: The dynamics include a consensus kernel, which draws neighbors in $z$-space closer together. We see that the population moves to make the classifier performer better, as the two distributions become more easily separable by the linear classifier.

In Figure 4, we observe the strategic distribution separating itself from the stationary distribution, improving the performance of the classifier and also improving the performance of the population itself. The strategic distribution and classifier appear to be stationary by time $t = 40$.

## E.2  Multiple Dynamical Populations

We also want to understand the dynamics when both populations are strategic and respond to the classifier. In this example, we numerically simulate this and in future work we hope to prove additional results regarding convergence. This corresponds to modeling the previously-fixed distribution $\bar{\rho}$ as time-dependent; let this distribution be $\tau \in \mathcal{P}_2$. We consider the case where $\rho$ is competitive with $x$ and $\tau$ is aligned with $x$, with dynamics given by

$$\partial_t \rho = -\text{div}\left(\rho \nabla_z \left(f_1(z, x) - \alpha \log(\rho/\tilde{\rho}) - W * \rho\right)\right)$$
$$\partial_t \tau = \text{div}\left(\tau \nabla_z \left(f_2(z, x) + \alpha \log(\tau/\tilde{\tau}) + W * \tau\right)\right)$$
$$\frac{\mathrm{d}}{\mathrm{d}t} x = -\nabla_x \left(\int f_1(z, x)\mathrm{d}\rho(z) + \int f_2(z, x)\mathrm{d}\tau(z) + \frac{\beta}{2}\|x - x_0\|^2\right).$$

We use $W = 0$ and $f_1$, $f_2$ as in section 4.1 and the same discretization parameters as in Section E.1. In Figure 5,

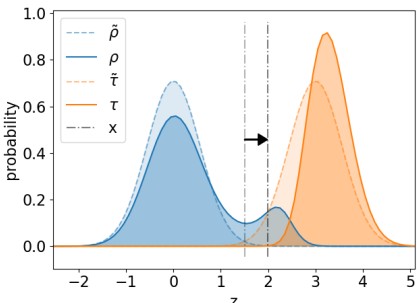

Figure 5: The population $\rho$ aims to be classified with the $\tau$ population, while the classifier moves to delineate between the two. We observe that $\tau$ adjusts to improve the performance of the classifier while $\rho$ competes against it. The distributions are plotted at time $t = 0$, corresponding to $\tilde{\rho}$ and $\tilde{\tau}$, and time $t = 20$, corresponding to $\rho$ and $\tau$.

we observe that the $\tau$ population moves to the right, assisting the classifier in maintaining accurate scoring. In contrast, $\rho$ also moves to the right, rendering the right tail to be classified incorrectly, which is desirable for individuals in the $\rho$ population but not desirable for the classifier. While we leave analyzing the long-term behavior mathematically for future work, the distributions and classifier appear to converge by time $t = 20$.

## E.3  Sampled Gradients

In real-world applications of classifiers, the algorithm may not know the exact distribution of the population, relying on sampling to estimate it. In this section we explore the effects of the classifier updating based on an approximated gradient, which is computed by sampling the true underlying distributions $\rho$ and $\bar{\rho}$. We use the same parameters for the population dynamics as in section 4.1, and for the classifier we use the approximate gradient

$$\nabla_x L(z, x_t) \approx \frac{1}{n}\sum_{i=1}^{n} \left(\nabla_x f_1(z_i, x_t) + \nabla_x f_2(\bar{z}_i, x_t)\right) + \beta(x_t - x_0), \quad z_i \sim \rho_t, \quad \bar{z}_i \sim \bar{\rho}_t.$$

First, we simulate the dynamics with the classifier and the strategic population updating at the same rate, using $\alpha = 0.05$, $\beta = 1$, and the same consensus kernel as used previously, with the same discretization parameters as in E.1. In Figure 6, we observe no visual difference between the two results with $n = 4$ versus $n = 40$ samples, which suggests that not many samples are needed to estimate the gradient.

Next, we consider the setting where the classifier is best-responding to the strategic population.

Unlike the first setting, we observe in Figure 7 a noticeable difference between the evolution of $\rho_t$ with $n = 4$ versus $n = 40$ samples. This is not surprising because optimizing with a very poor estimate of the cost function at each time step would cause $x_t$ to vary wildly, and this method fails to take advantage of correct "average" behavior that gradient descent provides.

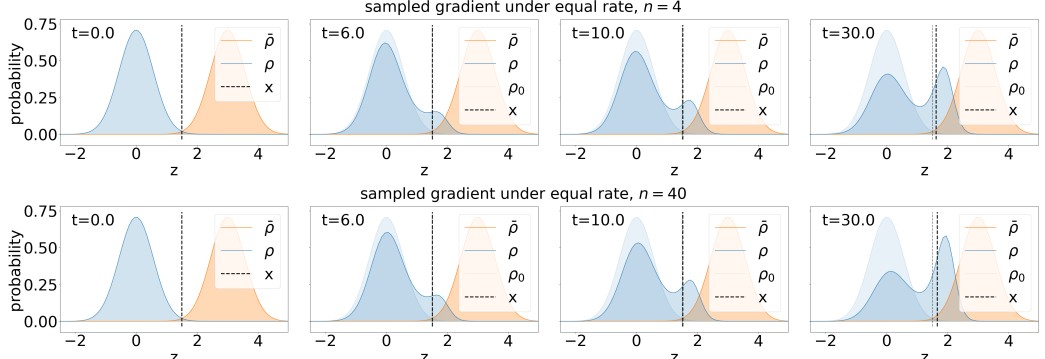

Figure 6: When the classifier is updating at the same rate as the population, we do not see a significant change in the evolution of both species, suggesting that as long as the gradient estimate for the classifier is correct on average, the estimate itself does not need to be particularly accurate.

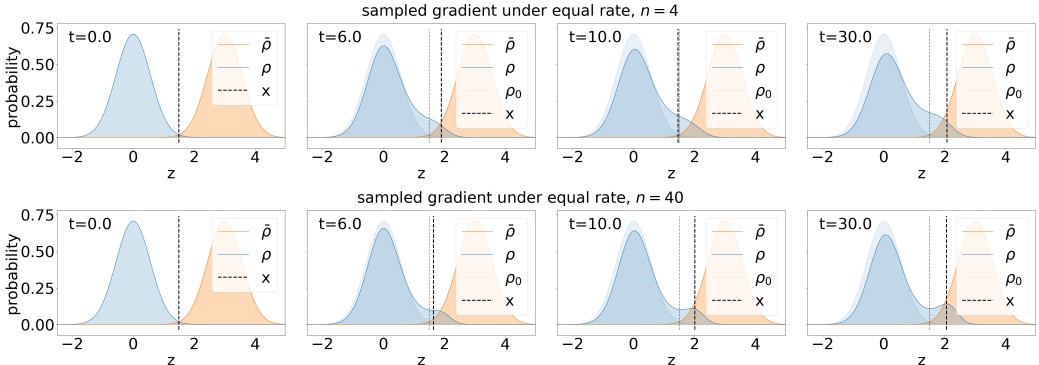

Figure 7: When the classifier is best-responding to the population, we observe that using $n = 4$ samples leads to different behavior for both the classifier and the population, compared with a more accurate estimate using $n = 40$ samples.

### E.4 Two-dimensional Distributions

In practice, individuals may alter more that one of their attributes in response to an algorithm, for example, both cancelling a credit card and also reporting a different income in an effort to change a credit score. We model this case with $z \in \mathbb{R}^2$ and $x \in \mathbb{R}^2$, and simulate the results for the setting where the classifier and the population are evolving at the same rate. While this setting is not covered in our theory, it interpolates between the two timescale extremes.

We consider the following classifier:

$$
\begin{aligned}
f_1(z, x) &= \frac{1}{2}\left(1 - \frac{1}{1 + \exp x^\top z}\right) \\
f_2(z, x) &= \frac{1}{2}\left(\frac{1}{1 + \exp x^\top z}\right)
\end{aligned}
\tag{41}
$$

with $W = 0$. Again, the reference distribution $\tilde{\rho}$ corresponds to the initial shape of the distribution, instituting a penalty for deviating from the initial distribution. We use $\alpha = 0.5$ and $\beta = 1$ for the penalty weights, run for $t = 4$ with $dt = 0.005$ and $dx = dy = 0.2$ for the discretization. In this case, the strategic population is competing with the classifier, with dynamics given by

$$
\begin{aligned}
\partial_t \rho &= -\operatorname{div}\left(\rho \nabla_z \left(f_1(z, x) - \alpha \log(\rho/\tilde{\rho})\right)\right) \\
\frac{\mathrm{d}}{\mathrm{d}t} x &= -\nabla_x \left(\int f_1(z, x) \mathrm{d}\rho(z) + \int f_2(z, x) \mathrm{d}\bar{\rho}(z) + \frac{\beta}{2} \|x - x_0\|^2\right)
\end{aligned}
$$

In Figure 8, we observe the strategic population increasing mass toward the region of higher probability of being labeled "1" while the true underlying label is zero, with the probability plotted at time $t = 4$. This illustrates

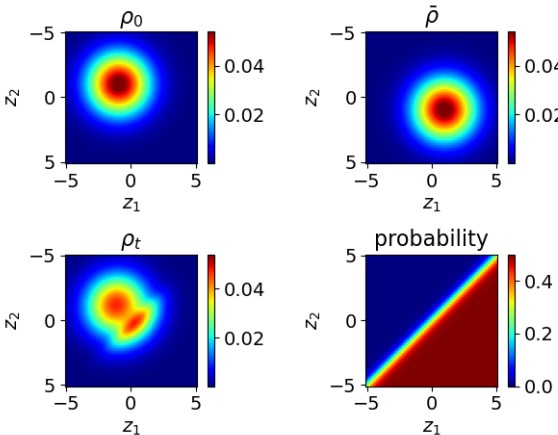

Figure 8: We use (41) for the classifier functions, using a Gaussian initial condition and regularizer for $\rho$. We see the distribution moving toward the region with higher probability of misclassification.

similar behavior to the one-dimensional case, including the distribution splitting into two modes, which is another example of polarization induced by the classifier. Note that while in this example, $x \in \mathbb{R}^2$ and we use a linear classifier; we could have $x \in \mathbb{R}^d$ with $d > 2$ and different functions for $f_1$ and $f_2$ which yield a nonlinear classifier; our theory in the timescale-separated case holds as long as the convexity and smoothness assumptions on $f_1$ and $f_2$ are satisfied.

