# OpenReview forum: "Strategic Distribution Shift of Interacting Agents via Coupled Gradient Flows"
_NeurIPS.cc/2023/Conference — NeurIPS 2023 poster_

### Official Review · Reviewer_nK6f · 2023-07-04

**Soundness:** 4 excellent
**Presentation:** 3 good
**Contribution:** 4 excellent
**Rating:** 7
**Confidence:** 3

**Summary:**

The authors consider a the problem where the change in a distribution for an objective can be modeled as a set of coupled nonlinear parabolic PDEs. These methods have nonlocal interactions and describe a model of the influence of the model on the population and vice-versa. Additionally, these correspond to the evolution of the measures associated with the Wasserstein gradient flow that minimizes a set of defined energy functionals, which relate to the original optimization problem. Such equations have been well studied in terms of the granular media equation and a wide theory has been devoted to existence and uniqueness of their solution. Two scenarios are considered: one with cooperation and another with an adversarial behavior of the population. The main results of the paper are convergence rates to the steady state for the coupled PDEs. The behavior of both the optimization algorithm and the population are studied in some numerical experiments at the end of the paper. The numerical experiments illustrate the importance of considering such a model rather than relying on more simple summary statistics for notions of distributional shift.

**Strengths:**

The proposed model is a very elegant model for describing distribution shift and the feedback mechanisms between the shift in population and objective functions. It provides a much richer class of perturbations than what is generally seen in the literature in cases such as distributionally robust optimization or in adversarial optimization. The model also has implications in providing additional distribution information (beyond low order moments) at equilibrium. The authors show the importance of this where different geometries of the distribution can imply different effects on the population distribution. Overall, the paper is well written, provides nice ideas, and describes a more unique take on the problem of distribution shifts.

**Weaknesses:**

The numerical evaluation is a bit limited, but I think that’s not a problem since most of the results are theoretical. Still, it would be nice to see some more concrete applications of the theory by simulating the PDEs described, where, for example, all components are not simulated. It also appears that this may be difficult to apply to real world scenarios as the authors alluded in their limitations. A real application where all components need to be estimated was not discussed, but that's beyond the scope of the paper.

Some of the assumptions are somewhat strong, but these are usually made to provide convergence and existence of solutions of these PDEs. In that regard, processes that satisfy these assumptions may not be entirely general, but that’s not a big deal since the authors clearly constrain their analysis based on the assumptions they make.




**Questions:**

The second equation should be the argmin of x \in R^d?

Is there intuition on how the different functionals could be estimated e.g. for a particular application? Or is it assumed that these are already known?

Related to the previous question, how difficult would it be to apply to a real scenario, beyond the simulations the authors provided? This may not be a focus of the paper, but seems like something worth discussing.

Could particle methods be used to solve the problem in high dimensions as is sometimes done in high dimensional cases? For example, this was explored in [1] and I was wondering if it could be applied in this scenario by approximating the associated McKean-Vlasov process (there might be an issue with some of the terms in the PDEs)?

[1] Crucinio et al, Solving Fredholm Integral Equations of the First Kind via Wasserstein Gradient Flows, 2022

**Limitations:**

The authors sufficiently discuss limitations of their work.

---

> ### Author Rebuttal · Authors · 2023-08-09
>
> Thank you for your detailed, specific, and very helpful feedback on our work.
>
> **Question 1:** Thank you for catching this; we had a typo. The equation should be $$\underset{z\in\mathbb{R}^d}{\operatorname{argmin}}J(z,x)$$ because this is modeling an individual agent, whose data is given by $z$, optimizing their own utility with respect to given algorithm parameters $x\in\mathbb{R}^d$, whereas the first equation represents the algorithm optimizing a loss over the entire population of agents.
>
> **Question 2:** This is a great idea; we are really interested and excited about investigating this in our future work. We have been considering  kernel methods and Gaussian processes to accomplish this task in addition to machine learning approaches.
>
> **Question 3:** To illustrate the relevance of this new framework, we show how our model can be used to fit real-world data. In Figure 1 [see attached PDF], poverty index scores collected in Colombia [CC11] are shown before and after the threshold for qualifying for government aid (shown in red) was released. We observe that many people shifted their data to be just under the threshold. We can model this with our PDE, as shown in Figure 2, using basic intuition to fit the PDE. We will add this to our paper.
>
> **Question 4:** Using stochastic differential equations (SDEs) as in the paper that was referenced would be a good way to simulate our PDEs in higher dimensions. Additionally, a number of recent works such as [Li+22,SS18] show promise in using deep learning to solve PDEs in high dimensions. As a subcase of the high-dimensional setting, we would like to investigate settings where agents are resource bound and can only perturb one or two features in a larger set of features, since in reality humans have limited resources for feature manipulation.
>
> **Note on additional figure:**
> In comparison with prior work that addresses learning in the face of distribution shift, we consider the methods used in [MPZ21], wherein the learning algorithm fits a linear mean-shift model to the algorithm parameters. Our framework provides a method for comparing the outcome of using a mean-shift model versus continually retraining the algorithm via gradient descent, which is shown in Figure 3. This simulation shows that state-of-the-art techniques for responding to distribution shift can perform poorly compared to naive techniques, emphasizing the need for better modeling. The framework itself is novel as PDEs have not been used to model distribution shift in the context of people interacting with algorithms, and takes a similar form as PDEs which model populations of animals and bacteria.
>
> [CC11] Adriana Camacho and Emily Conover. “Manipulation of Social Program Eligibility”. In: American Economic
> Journal: Economic Policy 3.2 (2011), pp. 41–65. issn: 1945-7731.
>
> [SS18] Justin Sirignano and Konstantinos Spiliopoulos. “DGM: A deep learning algorithm for solving partial differential
> equations”. In: Journal of Computational Physics 375 (Dec. 2018), pp. 1339–1364.
>
> [MPZ21] John P. Miller, Juan C. Perdomo, and Tijana Zrnic. “Outside the Echo Chamber: Optimizing the Performa-
> tive Risk”. In: Proceedings of the 38th International Conference on Machine Learning. International Conference
> on Machine Learning.
>
> [Li+22] Zongyi Li et al. Fourier Neural Operator with Learned Deformations for PDEs on General Geometries.

---

> > ### Comment · Reviewer_nK6f · 2023-08-20
> >
> > Thank you very much to the authors for the response. I enjoyed this paper and I will continue recommending its acceptance.

---

### Official Review · Reviewer_vBit · 2023-07-07

**Soundness:** 3 good
**Presentation:** 3 good
**Contribution:** 2 fair
**Rating:** 7
**Confidence:** 2

**Summary:**

This paper studies the long-term behavior of a dynamical system where there are distribution shifts in response to algorithmic decision-making. The authors derive PDEs to capture distributional changes and describe the result of the algorithm for both the coorperative setting and the competitive setting. Under certain regularization conditions, it is shown that the PDEs converge to unique steady-state distribution. Finally, some numerical simulations are conducted to iilustrate the problem.

**Strengths:**

1. The paper is well-written and not hard to read. All definitions, theoretical results and experiments are accompanied by explanations when necessary. Moreover, the authors motivate the problem of interest in a very convincing way.

2. The paper is technically solid. The process of interaction between algorithm and population is rigorously modelled and its asymptotic behavior is characterized.

**Weaknesses:**

The authors do not include many discussions on the insights provided by the theoretical findings. This is probably because the limit distribution of the PDEs are difficult to characterize. Moreover, the setting considered is over-simplified since it only contains one agent. Given that this paper is kind of a first step towards mathematical analysis of algorithm interaction, I think these flaws are totally acceptable and can be left for future work.

**Questions:**

1. Many real-world applications involve discrete rather than continuous time. Is it possible to generalize the analysis in this paper to discrete time setting?

2. Is it reasonable to assume that $f_1,f_2$ are convex?

**Limitations:**

The authors have adequately addressed the limitations and potential negative societal impact of their work.

---

> ### Author Rebuttal · Authors · 2023-08-09
>
> Thank you for your helpful feedback and questions.
>
> **Weaknesses:** We agree that more of a discussion about insights would improve our paper and we can add this in the next version. One key insight is the identification of a large classes of input functions that lead to exponential convergence with explicit rates that can be computed directly from model parameters, telling us how fast we can expect the dynamics to converge. Another key insight is the potential of this PDE formulation to allow for methodology transfer from other application fields such as math-biology and physics in terms of the analysis tools used for predicting the behavior of solutions.
> We are excited about extending this framework beyond one type of agent (note that currently we have an infinite number agents, same type) in our next iteration, where we would have different "species". For instance, in our current setting $\bar \rho$ is a fixed reference measure; we expect our framework to be able to capture multi-species settings as well, such as the case when $\bar \rho$ also changes dynamically. We see no methodological limitations to extend our results to such multi-species systems in the future.
>
> **Question 1:** We could also consider these dynamics in discrete time and that is a good future direction. The goal of this paper is to analyze the precise distribution shift from a continuous optimization perspective. Any results achieved for the continuum dynamics directly translate into approximate statements on the level of the time discretized analogues.
>
> **Question 2:** We assumed that $f_1$ and $f_2$ are convex for the purposes of the analysis, which covers many reasonable scenarios such as the log-loss, a commonly-used ML loss function. There are several ways how the assumptions could be relaxed, see for example the remarks in [CMV03]; we did not opt for the most general presentation of assumptions in this work as it would increase the technicality of the proofs without much conceptual gain for the presented frameworks, given that the current assumptions already cover large classes functions relevant in practice, including the prototypical example considered in Section 4. For instance, in [CMV03] the convexity of the potential term (corresponding to $f_1$ and $f_2$ in our case) can be relaxed and $\tilde\rho$ and/or $W$ can compensate to ensure convergence; similar generalizations could be applied in our setting.
>
> **Note on additional simulations:** In comparison with prior work that addresses learning in the face of distribution shift, we consider the methods used in [MPZ21], wherein the learning algorithm fits a linear mean-shift model to the algorithm parameters. Our framework provides a method for comparing the outcome of using a mean-shift model versus continually retraining the algorithm via gradient descent, which is shown in Figure 3 [see attached PDF]. This simulation shows that state-of-the-art techniques for responding to distribution shift can perform poorly compared to naive techniques, emphasizing the need for better modeling. The framework itself is novel as PDEs have not been used to model distribution shift in the context of people interacting with algorithms, and takes a similar form as PDEs which model populations of animals and bacteria.
>
> To illustrate the relevance of this new framework, we show how our model can be used to fit real-world data. In Figure 1, poverty index scores collected in Colombia [CC11] are shown before and after the threshold for qualifying for government aid (shown in red) was released. We observe that many people shifted their data to be just under the threshold. We can model this with our proposed PDE framework, as shown in Figure 2.
>
> [CMV03] Jose A. Carrillo, Robert J. McCann, and Cedric Villani. “Kinetic equilibration rates for granular media and related
> equations: entropy dissipation and mass transportation estimates”. In: Revista Matematica Iberoamericana 19.3
> (Dec. 2003), pp. 971–1018. issn: 0213-2230.
>
> [CC11] Adriana Camacho and Emily Conover. “Manipulation of Social Program Eligibility”. In: American Economic
> Journal: Economic Policy 3.2 (2011), pp. 41–65. issn: 1945-7731.
>
> [MPZ21] John P. Miller, Juan C. Perdomo, and Tijana Zrnic. “Outside the Echo Chamber: Optimizing the Performative Risk”. In: Proceedings of the 38th International Conference on Machine Learning. International Conference
> on Machine Learning.

---

> > ### Comment · Reviewer_vBit · 2023-08-15
> >
> > I would like to thank the authors for the detailed explanations and I will keep my rating.

---

### Official Review · Reviewer_HH7Z · 2023-07-08

**Soundness:** 3 good
**Presentation:** 3 good
**Contribution:** 2 fair
**Rating:** 5
**Confidence:** 2

**Summary:**

This paper presents a framework for analyzing dynamics of distribution shift that captures feedback loop between learning algorithms and the distributions that they are deployed on using a coupled differential equations model. The paper considers both co-operative and competitive settings and suggests that a learner updated with gradient descent converges to steady state both in finite and infinite dimensions (in terms of model parameters).

**Strengths:**

The paper presents a rigorous treatment of the problem and presents general results, although I am not an expert in this area and cannot present a rating with any meaningful confidence.

**Weaknesses:**

The paper works with several assumptions that I do not see a clear cut comparison of how these assumption compares against prior work, so it is unclear how general these results are.

**Questions:**

One way to make the paper appeal to a general audience is to have a discussion surrounding claims and assumptions comparing this work against prior results in the literature.

---

> ### Author Rebuttal · Authors · 2023-08-09
>
> Thank you for your review and feedback.
>
> **Questions:** In comparison with prior work that addresses learning in the face of distribution shift, we can analyze with the methods used in [MPZ21], wherein the learning algorithm fits a linear mean-shift model to the algorithm parameters. Our framework provides a method for comparing the outcome of using a mean-shift model versus continually retraining the algorithm via gradient descent, which is shown in Figure 3. This simulation shows that state-of-the-art techniques for responding to distribution shift can perform poorly compared to naive techniques, emphasizing the need for better modeling. The framework itself is novel as PDEs have not been used to model distribution shift in the context of people interacting with algorithms, and takes a similar form as PDEs which model populations of animals and bacteria. We will add this to the paper to provide more context for our framework.
>
> We will add more details on motivation and justification of our assumptions, extending the top of page 6 in our paper where we have examples of functions which satisfy the assumptions, including the log-loss function which is commonly used in machine learning problems. More broadly, the assumptions on the input functions are commonly used and well-studied for related classes of PDE models. There are several ways how they could be relaxed, see for example the remarks in [CMV03]; we did not opt for the most general presentation of assumptions in this work as it would increase the technicality of the proofs without much conceptual gain for the presented frameworks, given that the current assumptions already cover large classes functions relevant in practice, including the prototypical example considered in Section 4.
>
> To illustrate the relevance of this new framework, we show how our model can be used to fit real-world data. In Figure 1 [see the attached PDF], poverty index scores collected in Colombia [CC11] are shown before and after the threshold for qualifying for government aid (shown in red) was released. We observe that many people shifted their data to be just under the threshold. We can model this with our PDE, as shown in Figure 2. We will also add this to the paper.
>
> Is there any other information we could provide to help improve our paper?
>
> [CMV03] Jose A. Carrillo, Robert J. McCann, and Cedric Villani. “Kinetic equilibration rates for granular media and related equations: entropy dissipation and mass transportation estimates”. In: Revista Matematica Iberoamericana 19.3 (Dec. 2003), pp. 971–1018. issn: 0213-2230.
>
> [MPZ21] John P. Miller, Juan C. Perdomo, and Tijana Zrnic. “Outside the Echo Chamber: Optimizing the Performative Risk”. In: Proceedings of the 38th International Conference on Machine Learning. International Conference
> on Machine Learning.
>
> [CC11] Adriana Camacho and Emily Conover. “Manipulation of Social Program Eligibility”. In: American Economic
> Journal: Economic Policy 3.2 (2011), pp. 41–65. issn: 1945-7731.

---

> > ### Comment · Reviewer_HH7Z · 2023-08-18
> > **Re. author response**
> >
> > Thanks to the authors for their response. I do not have any other questions, and will retain my score as is.

---

### Official Review · Reviewer_P1kC · 2023-07-26

**Soundness:** 3 good
**Presentation:** 3 good
**Contribution:** 3 good
**Rating:** 6
**Confidence:** 3

**Summary:**

The paper proposes a new framework for analyzing the interaction and feedback loop between learning algorithms and the distributions on which they are deployed. The framework in the paper is more general than prior models, allowing to capture more complex interactions and shifts in the algorithm and distribution. This is done using a partial differential equation that modifies the algorithm\distribution in a way that optimizes its cost with respect to the choice of distribution\algorithm. Under several assumptions which include strong convexity, the authors were able to prove that gradient flow converges and provide fast convergence rates. The authors are provide experimental results for a simple 1-dimensional setting which demonstrate that their framework can explain general phenomenon which are not explained by prior models, such as polarization.

**Strengths:**

The paper is well-written and easy to follow. Moreover, as mentioned above, the framework in the paper offers several advantages compared to prior frameworks, allowing to capture more complex interactions and shifts of the distribution. This model is therefore able (via simple examples) to theoretically explain forms of distribution shifts which we see in practice such as polarization.


**Weaknesses:**


1. While the framework allows for complex forms of shifts of the model and distribution, the analysis assumes that the model\distribution changes to be the exact minimizer of a certain cost function. This may be a little strict in general. Can the same analysis work when the shifts are approximate minimizers for the cost function?

2. Extreme timescale setting: the analysis in the paper only considers two extreme timescale settings: (i) the algorithm responds much faster than the population, and (ii) the population responds much faster than the algorithm. This is somewhat strict and doesn’t allow for intermediate changes in the algorithm\distribution.

3. Assumptions: the paper has many assumptions, some of which are less common. The authors should give some motivation and justification for their assumptions, and provide examples where all of these assumptions hold.

4. The experiments are a little simple (1-dimensional classifiers) and it would be more interesting to see an example which is more relevant in practice. This is perhaps one of the limitations of this framework: even though it is able to capture more complex structures of shifts, it is harder to simulate for more sophisticated examples and models (e.g. high dimensional classifiers, other timescale settings, and so on). The authors should therefore also discuss the limitation of their framework compared to existing ones.

5. Experimental discussion: the experimental section jumps too fast to the experiments without explaining the objectives of these experiments or discussing its connections to the previous theoretical results. The authors should explain their main objectives in this setting, discuss connection to the theory in the previous section, and also compare to existing prior work (e.g. show how the example in section 4.1 cannot be captured by prior frameworks).


**Questions:**

See above.

**Limitations:**

See above.

---

> ### Author Rebuttal · Authors · 2023-08-09
>
> Thank you for your detailed response and helpful comments.
>
> **Model Optimization:** The setting where the dynamics are not exactly minimizing a cost functional is an interesting idea for future work; our work can be seen as a necessary first step for building more general frameworks in this direction. While we considered the case where the distribution evolves towards the exact minimizer of a given cost function, in practice, this cost function would be approximated from data resulting in a shift that is not exactly solving the true minimization problem. We also note that for the gradient flow formulation, the minimizer is reached as time goes to infinity. In practice, the evolution would be solved up to finite time only, therefore providing an approximation of the true minimizer. By choosing this final time large enough, arbitrary accuracy can be achieved.  The convergence rates we obtain provide quantitative error bounds both in energy and in Wasserstein distance that measure how close the shift obtained at any chosen final time is to the true minimizer.
>
> **Extreme timescale setting:** While we provide numerical examples of the intermediate timescale separation case, we leave rigorous analysis for future work. Different analytical tools are likely needed for a rigorous treatment of this case.
>
> **Assumptions:** We will add more details on motivation and justification of these assumptions, extending the top of page 6 in our paper where we have examples of functions which satisfy the assumptions, including the log-loss function which is commonly used in machine learning problems.
>
> **Regarding Limitations:** We will add a limitations section highlighting scaling with dimension as that is a key challenge for our framework. Besides using SDEs and deep learning to solve these systems of PDEs in higher dimensions, we would like to investigate settings where agents are resource bound and can only perturb one or two features in a larger set of features, since in reality humans have limited resources for feature manipulation.
>
> In comparison with prior work that addresses learning in the face of distribution shift, consider the methods used in [MPZ21], wherein the learning algorithm fits a linear mean-shift model to the algorithm parameters. Our framework provides a method for comparing the outcome of using a mean-shift model versus continually retraining the algorithm via gradient descent, which is shown in Figure 3. This simulation shows that state-of-the-art techniques for responding to distribution shift can perform poorly compared to naive techniques, emphasizing the need for better modeling. The framework itself is novel as PDEs have not been used to model distribution shift in the context of people interacting with algorithms, and takes a similar form as PDEs which model populations of animals and bacteria.
>
> To illustrate the relevance of this new framework, we show how our model can be used to fit real-world data. In Figure 1 [see the attached PDF], poverty index scores collected in Colombia [CC11] are shown before and after the threshold for qualifying for government aid (shown in red) was released. We observe that many people shifted their data to be just under the threshold. We can model this with our proposed PDE framework, as shown in Figure 2. We will add both of these examples to our paper.
>
> **Experimental Discussion:** We will provide more context and explain our experiments more clearly; thank you for the suggestion. Other than the 2d classifier we simulated, we leave implementations in higher dimensions for future work. Using stochastic differential equations (SDEs) would be a reasonable strategy for simulations in higher dimensions. Additionally, a number of recent works such as [Li+22,SS18] show promise in using deep learning to solve PDEs in high dimensions.
>
> [CC11] Adriana Camacho and Emily Conover. “Manipulation of Social Program Eligibility”. In: American Economic
> Journal: Economic Policy 3.2 (2011), pp. 41–65. issn: 1945-7731.
>
> [SS18] Justin Sirignano and Konstantinos Spiliopoulos. “DGM: A deep learning algorithm for solving partial differential
> equations”. In: Journal of Computational Physics 375 (Dec. 2018), pp. 1339–1364.
>
> [MPZ21] John P. Miller, Juan C. Perdomo, and Tijana Zrnic. “Outside the Echo Chamber: Optimizing the Performa-
> tive Risk”. In: Proceedings of the 38th International Conference on Machine Learning. International Conference
> on Machine Learning.
>
> [Li+22] Zongyi Li et al. Fourier Neural Operator with Learned Deformations for PDEs on General Geometries.

---

> > ### Comment · Reviewer_P1kC · 2023-08-14
> > **Response**
> >
> > Thanks for the detailed response; I'm still recommending to accept the paper.

---

### Author Rebuttal · Authors · 2023-08-09

See figures in attachment.

---

### Decision · Program_Chairs · 2023-09-21

**Decision:**

Accept (poster)

**Comment:**

The paper proposes a novel framework for analyzing distribution shift and feedback mechanisms between learning algorithms and the distributions they operate on using coupled nonlinear parabolic partial differential equations (PDEs). The reviews generally found the paper technically sound and appreciated its contribution. The reviewers noted the elegance of the model and its potential to capture complex interactions in distribution shifts as well as a richer class of perturbations compared to existing literature.

However, there are some concerns and suggestions raised by the reviewers:

1. Reviewers suggest that the paper could benefit from a more extensive discussion of insights derived from the theoretical findings. They recommend adding more context about the practical implications and applications of the model.
2. While the paper is primarily theoretical, some reviewers mention that additional numerical experiments could enhance its relevance. This could include simulating the PDEs described in various scenarios or providing more concrete examples of how the framework can be applied.
3. Some reviewers note that the assumptions made in the paper are strong, but they acknowledge the need for such assumptions to ensure convergence and existence of solutions for the PDEs. The authors should further clarify and justify these assumptions.
4. Reviewers express interest in understanding how challenging it would be to apply the framework in real-world scenarios. While the limitations are discussed, providing more insights into the practical challenges and potential applications could be valuable.

Overall, the paper was well-received, and the reviewers commended the authors for their detailed responses to the feedback. Addressing the suggestions and adding more context and insights to the paper could further enhance its impact and practical relevance. Therefore, I recommend accepting the paper with revisions based on the feedback provided.